# Adversarial Training for Process Reward Models

**Gurusha Juneja** [1]  **Deepak Nathani** [1]  **William Yang Wang** [1]

## Abstract

Process Reward Models (PRMs) enhance reasoning ability of LLMs by providing step-level supervision. However, their widespread adoption is limited due to expensive manual step-level annotation and poor generalization of static training data to novel errors. We introduce Adversarially Trained PRMs (APRM), where a Generator ($G$) learns to produce reasoning errors to deceive a PRM ($R$), while $R$ concurrently learns to detect them. This interaction yields progressively harder negatives for $R$, improving it's robustness and generalization to novel errors without requiring manual step-level labels. Averaged across diverse mathematical reasoning benchmarks, APRM improves solver accuracy by $+3.4$ percentage points (pp) over the strongest PRM baseline. APRM achieves gains of $+5.3$ pp on out-of-distribution tasks.

*"It is not the strongest of the species that survives, nor the most intelligent, but the one most adaptable to change"*

*- Charles Darwin*

## 1. Introduction

Large Language Models (LLMs) have demonstrated impressive capabilities in complex reasoning, even achieving gold-medal-level performance on the International Mathematical Olympiad questions, when augmented with specialized systems (Trinh et al., 2024; Luong et al., 2025). However, their reasoning remains fundamentally unreliable, often producing factually incorrect outputs despite appearing coherent (Amini & Kosseim, 2022; Bommasani, 2021; Uesato et al., 2022). This unreliability poses significant risks, particularly in high-stakes domains such as scientific discovery, finance, or medicine, where subtle errors can have severe consequences. As these models become better, their errors become more nuanced (Lightman et al., 2023c;

Zhang et al., 2024; Guan et al., 2025; Achiam et al., 2023). Detecting such errors is essential for trustworthy reasoning, necessitating the need for Process Reward Models (PRMs) that can identifying very subtle errors.

Current PRM training techniques predominantly rely on static datasets (Lightman et al., 2023a; Wang et al., 2024a; Lightman et al., 2023b). These methods are inherently limited as they provide a fixed error distribution that cannot adapt to more nuanced errors. On the other hand, synthetic data generation techniques like (Zhang et al., 2024; Lu et al., 2024b; Cobbe et al., 2021; Yu et al., 2024b; Uesato et al., 2022) rely on the flawed assumption that correct final answers imply correct intermediate steps. These paradigms lack mechanisms to actively mine harder negatives.

To address this, we require a PRM training paradigm that provides an adaptive curriculum, where negative sample hardness dynamically increases while aligning with the PRM's evolving capabilities. This makes the PRM robust to increasingly subtle errors and also ensures optimal and smooth learning. In this paper, we introduce Adversarially Trained Process Reward Models (APRM)[1]. We formulate PRM training as a two-player, general-sum, non-cooperative game. In APRM (Fig 1), a Generator ($G$) actively learns to produce plausible but incorrect reasoning steps to deceive a Reward Model ($R$), which concurrently learns to detect these increasingly subtle errors. This competitive interaction creates the desired adaptive curriculum, directly addressing the limitations of static or passively generated training data.

APRM is both theoretically and empirically grounded. We employ game-aware optimizers and symmetric policy regularization to ensure stable training dynamics with provable linear-rate convergence to a Nash Equilibrium. Unlike prior adversarial approaches (Goodfellow et al., 2014; Madry et al., 2018; Wu et al., 2024) that were framed as zero-sum games, we model the Generator–PRM interaction as a general-sum game. This yields a stationary-point Nash Equilibrium rather than a mini-max saddle point. Experiments on mathematical reasoning benchmarks show that APRM outperforms state-of-the-art PRM training and prompting methods by up to **5.3 pp**, with the largest gains on out-of-

[1]University of California, Santa Barbara. Correspondence to: Gurusha Juneja <gurusha@ucsb.edu>.

*Proceedings of the 43rd International Conference on Machine Learning*, Seoul, South Korea. PMLR 306, 2026. Copyright 2026 by the author(s).

[1]Our code is available at `https://github.com/UCSB-AI/Adversarial-Process-Reward-Models`.

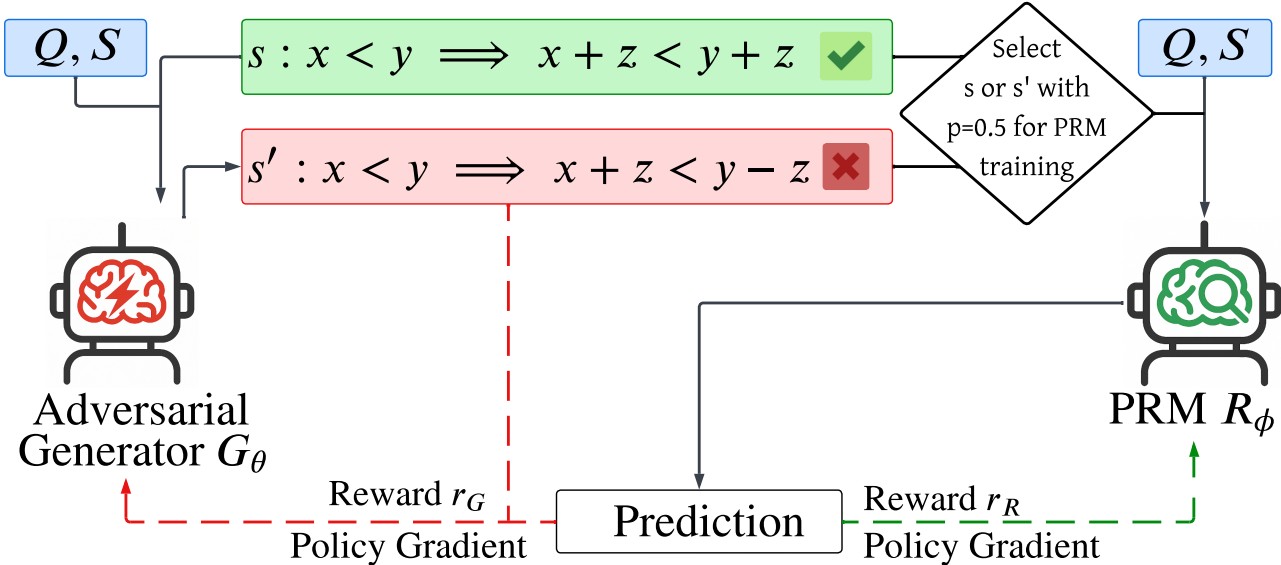

*Figure 1.* **APRM Overview.** The Generator $G_\theta$ perturbs the correct step $s$ into a plausible but incorrect intermediate reasoning step $s'$. While the PRM $R_\phi$ evaluates a step for it's correctness given the question $Q$ and partial solution $S$. $R_\phi$ is rewarded for correctly distinguishing between $s$ and $s'$, whereas, $G_\theta$ is trained to generate incorrect step. Both $G_\theta$ and $R_\phi$ receive rewards based on their prediction. Training alternates between improving $R_\phi$ (more robust detection of flawed reasoning) and improving $G_\theta$ (harder, more realistic negatives). further details in section 2.1.

distribution problems, highlighting it's enhanced robustness and generalization.

We summarize our contribution as follows:

- **Adversarial training for PRMs.** We introduce PRM training as a two-player adversarial game between a generator and a PRM. The generator learns to produce harder negatives, leading to stronger and more generalizable Process Reward Models.
- **Theoretical Analysis.** We provide theoretical analysis of the game, showing the existence of Nash Equilibria. We establish linear convergence guarantees under regularized game utilities and use game-aware optimizers to enable stable training.
- **Experimental Validation.** We experimentally validate the proposed adversarial training setup. Across math and science reasoning benchmarks, APRM yields an average improvement of **3.4 pp** over state-of-the-art PRM training methods, consistently observed across solvers of different sizes.

## 2. Adversarial Process Reward Models

We formulate the problem of learning a robust process reward model (PRM) as a two-player adversarial game. This game consists of a Generator $G$, tasked with perturbing a correct reasoning steps to make it incorrect, and a Reward Model $R$, whose objective is to predict the correctness of a given reasoning step. G aims to deceive R into classifying the corrupted step as correct. R, on the other hand, aims for accurate classification, detecting even the most nuanced

errors generated by $G$ (Figure 1). Through this iterative competition, $G$ learns to generate increasingly subtle errors, while $R$ concurrently evolves to detect even the most nuanced flaws.

**Notation.** We denote the Generator by $G_\theta$ and the Reward Model by $R_\phi$, parameterized by weights $\theta \in \Theta$ and $\phi \in \Phi$, respectively. A correct reasoning step is represented as a sequence $s = \{s_1, s_2, \ldots, s_m\}$, where each $s_i$ is a token from the finite vocabulary $\mathcal{V}$.

The Generators action is a token sequence $a_G \in \mathcal{A}_G$, where $a_G = \{s'_1, s'_2, \ldots, s'_n\}$ with $s'_i \in \mathcal{V}$ and $n \leq \ell$ ($\ell$ is the maximum sequence length). The action space $\mathcal{A}_G$ thus consists of all token sequences of length 1 to $\ell$. These generated sequences correspond to corrupted reasoning steps.

The Reward Model's action is a binary classification $a_R \in \mathcal{A}_R = \{0, 1\}$, where 1 denotes Correct and 0 denotes Incorrect. The players use mixed strategies(policies) defined as $\pi_\theta \in \Delta(\mathcal{A}_G)$ for the Generator and $\pi_\phi \in \Delta(\mathcal{A}_R)$ for the Reward Model, where $\Delta(\cdot)$ denotes a probability distribution over the respective action spaces. Note that these strategies are distinct from the underlying model parameters $\theta$ and $\phi$.

Let $y(a_G) \in \{0, 1\}$ be the ground-truth correctness label of a reasoning step generated by $G_\theta$, i.e. $y = 1$ for a correct generated step and, 0 for incorrect generated step, $a_R = y'(a_G) \in \{0, 1\}$ be $R_\phi$'s classification output for $G$'s action. The scalar reward functions for $G_\theta$ and $R_\phi$ are represented by $r_G(y, y')$ and $r_R(y, y')$, jointly determined by the

ground-truth validity of $G_\theta$'s output and $R_\theta$'s classification. Detailed definitions are provided in Section 2.1

Utilities represent the expected payoffs for each player. Given a strategy profile $(\pi_\theta, \pi_\phi)$ and a distribution $p(s)$ over correct initial reasoning steps $s$, the utilities for both the players are given by:

$$U_G(\pi_\theta, \pi_\phi) = \mathbb{E}_{s \sim p(s), a_G, y' \sim (\pi_\theta, \pi_\phi)} \big[ r_G\big(y(a_G), y'\big) \big]$$
$$U_R(\pi_\theta, \pi_\phi) = \mathbb{E}_{s \sim p(s), a_G, y' \sim (\pi_\theta, \pi_\phi)} \big[ r_R\big(y(a_G), y'\big) \big] \quad (1)$$

### 2.1. Reward Function

We use reinforcement learning (RL) to train $G_\theta$ and $R_\phi$ through iterative adversarial interaction. The correctness $y(a_G)$ of $G$'s generation is approximated by an algorithmic proximity oracle [2] that uses non-LLM metrics, such as cosine similarity, entity matching (nouns and numbers), and logical operation order, relative to a reference solution. Further details of this oracle are provided in Appendix F. Given $y(a_G)$ and the Reward Model's classification $y'(a_G)$, the reward functions for both agents are structured as follows:

**Reward Function for $R_\phi$**  The PRM is rewarded for accurate classifications. It receives $+1$ for correct classifications $(y = y')$ and incurs a penalty of $-1$ for incorrect classifications $(y \neq y')$.

$$r_R(y, y') = \begin{cases} +1 & \text{if } y = y' \\ -1 & \text{if } y \neq y' \end{cases} \quad (2)$$

**Reward Function for $G_\theta$**  The Generator receives rewards that incentivize generation of deceptive errors. It is rewarded $+1$ for generating an incorrect reasoning step that successfully deceives $R$ into classifying it as correct $(y = 0$ and $y' = 1)$. It receives a reward of $-1$ for failing to corrupt a step (generating a correct step, $y = 1$) and $0$ for failing to decieve $R_\phi$ when it generated an incorrect step $(y = 0$ and $y' = 0)$.

$$r_G(y, y') = \begin{cases} +1 & \text{if } y = 0 \text{ and } y' = 1 \\ 0 & \text{if } y = 0 \text{ and } y' = 0 \\ -1 & \text{if } y = 1 \end{cases} \quad (3)$$

Note that since the reward/payoff functions of the two players are asymmetric (don't add up to the same value in every case), the game is formulated as a general-sum game.

### 2.2. Game Formulation and Equilibrium Concept

APRM formulates PRM training as a two-player, non-cooperative game, general-sum. The natural solution concept for this adversarial interaction is a Nash Equilibrium (Nash, 1951).

**Utility Functions.**  To ensure good optimization behavior (discussed in Section 2.3), our players optimize regularized utility functions. Both $G_\theta$'s and $R_\phi$'s utility is augmented with a Kullback-Leibler (KL) divergence penalty and an entropy bonus that help stabilize learning. These regularized utilities that define the precise objectives of the game, are defined as:

$$U'_G(\pi_\theta, \pi_\phi) = U_G(\pi_\theta, \pi_\phi) - \tau \text{KL}(\pi_\theta || \pi_{\theta_0}) + c_H H(\pi_\theta)$$
$$U'_R(\pi_\theta, \pi_\phi) = U_R(\pi_\theta, \pi_\phi) - \tau \text{KL}(\pi_\phi || \pi_{\phi_0}) + c_H H(\pi_\phi) \quad (4)$$

Where $U_G(\pi_\theta, \pi_\phi)$ and $U_R(\pi_\theta, \pi_\phi)$ are player utilities defined as expected payoffs(rewards) in Section 2. $\tau, c_H > 0$ are coefficients for KL-regularization and entropy bonuses, respectively. $\pi_{\theta_0}$ and $\pi_{\phi_0}$ are the policies of the initial pretrained models. See Appendix G for a discussion on mode collapse avoidance using these regularization.

**Theorem 1 (Existence of Nash Equilibrium)**  The adversarial game, characterized by agents $G_\theta$ and $R_\phi$, their regularized utility functions ($U'_G$ and $U'_R$), and action spaces ($\mathcal{A}_G$ and $\mathcal{A}_R$), is guaranteed to possess at least one Nash Equilibrium in mixed strategies.

**Proof**: We invoke *Glicksberg's generalization of Nash's existence theorem* (Glicksberg, 1952). The theorem guarantees the existence of a Nash Equilibrium in a game if two conditions are met:

1. *The players' strategy spaces are non-empty, convex, and compact*: Since the action spaces $\mathcal{A}_G, \mathcal{A}_R$ are finite and discrete, the corresponding policy spaces $\Delta(\mathcal{A}_G), \Delta(\mathcal{A}_R)$ are non-empty, convex, and compact (finite-dimensional simplices). [3]

2. *The players' utility functions are continuous in all players' strategies*: The utility functions $U'_G(\pi_\theta, \pi_\phi)$ and $U'_R(\pi_\theta, \pi_\phi)$ are expectations of bounded rewards under mixed (policy) distributions over actions. Because these expected-reward expressions are finite linear combinations (or integrals) of policy probabilities, they(including regularization) are continuous in the policy distributions $\pi_\theta$ and $\pi_\phi$.

---

[2] This oracle is not an absolute correctness checker, rather an approximation based on proximity of the generated step to the gold step and some symbolic checks. This cannot be used to check correctness of a standalone step during inference, hence, we do not use this as a baseline.

[3] The convexity referred to in Glicksberg's theorem is policy-space convexity (i.e. convexity of distributions over actions). This is different from convexity in the parameter space of a neural model (weights $\theta, \phi$), which is typically non-convex. Existence of an equilibrium in policy space does not imply that there is a single weight vector achieving that equilibrium, or that gradient-based training will find it. Nor do we claim in this paper that we are able to find the Nash equilibrium.

Given these conditions, Glicksberg's theorem ensures the existence of at least one Nash Equilibrium.

**Nash Equilibrium as a Stationary Point.** In a general-sum, non-cooperative games where the utilities are differentiable w.r.t policy parameter, a Nash Equilibrium is characterized as a *stationary point* in the joint strategy space (Raghunathan et al., 2019; Zhang et al., 2019). At such a point, no agent has a local incentive to unilaterally change their strategy. For APRM, this equilibrium signifies a state where $G_\theta$'s evolving ability to corrupt is optimally countered by $R_\phi$'s ability to detect those corruptions (given their capacities and the training data).

A *stationary point* is qualitatively different from the *saddle point* solution that frequently arises in works like Generative Adversarial Networks (Goodfellow et al., 2014) and robust optimization (Madry et al., 2018). These problems are formulated as zero-sum games where the equilibrium corresponds to a saddle point of a single objective function optimized by both players in perfectly opposing directions, contrary to our general sum formulation.

### 2.3. Convergence to Nash Equilibrium

In this section, we justify the introduction of regularization into the utility functions. The game described by the non-regularized utility functions $U_G$ and $U_R$ has non-monotone dynamics because the utilities are not simply negative correlated. In non-monotone settings, standard gradient methods often exhibit undesirable behaviors such as cycling, chaotic dynamics, or divergence, thus precluding direct convergence guarantees (Gidel et al., 2019). To mitigate these stability challenges and ensure provable convergence, we incorporate symmetric policy regularization into the player utility functions, as defined in Equation 4. Past works (Azar et al., 2024; Munos et al., 2024) prove that this regularization induces strong concavity in individual player objectives, which leads to a strong monotonicity and hence linear convergence to Nash Equilibrium.

**Lemma 1 (Strong Concavity from Regularization):** For any utility function $U(\pi)$ and a fixed reference policy $\pi_0$, the regularized objective $U'(\pi) = U(\pi) - \tau \cdot \text{KL}(\pi||\pi_0)$ is $\tau$-strongly concave with respect to the policy $\pi$. Similarly, an entropy bonus $\tau \cdot H(\pi)$ induces $\tau$-strong concavity (Azar et al., 2024).

Applying the lemma, and redefining the utility functions as given by Equation 4, we ensure that, $U'_G(\pi_\theta, \pi_\phi)$ and $U'_R(\pi_\theta, \pi_\phi)$ are $\tau$ and $c_H$-strongly concave with respect to the player policies. If each player's utility function is strongly concave with respect to its own policy, for any fixed opponent policy, then the operator $F(z) = (-\nabla(U_\theta(\pi_\theta, \pi_\phi)), -\nabla(U_\phi(\pi_\theta, \pi_\phi)))$, where $z = (\theta, \phi)$, becomes strongly monotone (Farnia & Ozdaglar, 2020).

Furthermore, for games characterized by a strongly monotone operator, game-aware first-order optimizers offer strong convergence guarantees, specifically ensuring linear convergence rates.

**Theorem 2 (Linear Convergence to Nash Equilibrium):** For the strongly monotone game described by the regularized utility functions (Section 2.2) algorithms such as Optimistic Gradient Descent-Ascent (OGDA) or Extra-Gradient (de Montbrun & Renault, 2022; Daskalakis et al., 2018) are guaranteed to generate a sequence of iterates $z_t$ that converges to the unique Nash Equilibrium $z^*$ at a linear rate. (For proof, see Appendix I).

### 2.4. Training Algorithm

Both $G_\theta$ and $R_\phi$ are trained using PPO (Schulman et al., 2017) augmented with Optimistic Gradient Descent-Ascent (OGDA) updates (see Section 2.3).

**Loss Function.** For each player, we augment the PPO clipped surrogate objective with KL and Entropy regularization terms. For instance, $G_\theta$'s the loss function (similarly for $R_\phi$) is given by:

$$L_G^{\text{total}}(\theta) = L_G^{\text{PPO,clip}}(\theta) + L_G^{\text{KL}}(\theta) + L_G^H(\theta) \quad (5)$$

Where minimizing $L^{PPO}$ approximately maximizes the un-regularized utility functions $U_G, U_R$ defined in Section 2. $L^{\text{KL}}$ given by $\tau \cdot \mathbb{E}_t [\text{KL}(\pi_\theta||\pi_{\theta_0})]$ and $L^H$ given by $-c_H \cdot \mathbb{E}_t [\text{H}(\pi_\theta)]$ correspond to the regularization introduced in Section 2.3.

**Game-Aware Optimization.** We replace the standard PPO's gradient updates by OGDA (Mertikopoulos et al., 2019; Daskalakis et al., 2018), which efficiently navigates the rotational game dynamics, in the regularized strongly monotone setting, by incorporating past gradient information. The update rule for OGDA is given by:

$$z_{t+1} = z_t - \eta(2F(z_t) - F(z_{t-1})) \quad (6)$$

where, $\eta$ is the learning rate, and $F(z_t)$ is is the gradient function w.r.t. the player's policy parameters, as described in section 2.3.

## 3. Experiments

### 3.1. Experimental Setting

**Model Architecture** Both the Generator ($G_\theta$) and the Process Reward Model ($R_\phi$) use `Llama-3.1-8B` as the backbone. This model offers a practical trade-off between computational cost and capacity for mathematical reasoning.

**Training** We train on the train split of MATH dataset (Hendrycks et al., 2021). The dataset provides with

*Table 1.* Performance comparison of APRM with the best MCTS based PRM training, adversarial training, hard negative based training and prompting methods on math reasoning benchmarks. Please see Table 9 for a full comparison with other PRM training baselines.

| Method | MATH500 | JEEB. | OlympiadB. | AIME25 | AMC | Avg. |
|---|---|---|---|---|---|---|
| **GPT-OSS-120B** | | | | | | |
| CoT-SC | 84.4 | 62.8 | 83.9 | 85.5 | 60.0 | 77.2 |
| LLM-J | 85.2 | 66.0 | 83.1 | 90.4 | **80.0** | 79.3 |
| ToT | 87.6 | 66.4 | 84.3 | 89.2 | 66.7 | 79.9 |
| ReST-MCTS | **91.6** | 69.1 | 84.3 | 91.0 | 67.4 | 81.7 |
| Hard Neg. | 91.2 | 69.5 | 79.2 | 91.5 | 68.5 | 80.0 |
| ARM | 89.2 | 62.4 | 77.6 | 87.9 | 63.3 | 76.1 |
| **APRM** (Ours) | 91.4 | **70.3** | **89.4** | **94.5** | 70.7 | **83.0** |
| **GPT-OSS-20B** | | | | | | |
| CoT-SC | 78.0 | 48.4 | 61.0 | 69.9 | 33.3 | 62.2 |
| LLM-J | 85.2 | 68.0 | 83.9 | 92.8 | 60.0 | 80.3 |
| ToT | 87.6 | 67.6 | 85.6 | 90.4 | 66.7 | 80.7 |
| ReST-MCTS | **91.0** | 68.3 | 84.7 | 91.3 | 66.0 | 82.3 |
| **APRM** (Ours) | **91.0** | **73.0** | **90.4** | 93.6 | **68.0** | **85.0** |
| **Gemma-3-27B** | | | | | | |
| CoT-SC | 78.8 | 48.8 | 48.7 | 69.9 | 13.3 | 58.4 |
| LLM-J | 86.4 | 69.2 | 83.1 | 91.6 | 73.3 | 80.4 |
| ToT | 88.0 | 67.6 | 85.2 | 88.0 | 60.0 | 80.2 |
| ReST-MCTS | **93.2** | 70.3 | 84.3 | 90.6 | **61.2** | 72.6 |
| Hard Neg. | 86.0 | 63.0 | 83.0 | 89.1 | 60.0 | 76.2 |
| ARM | 86.0 | 51.0 | 61.0 | 71.4 | 30.0 | 59.9 |
| **APRM** (Ours) | 91.4 | **74.4** | **90.7** | **91.9** | 60.8 | **85.2** |
| **Gemma-3-12B** | | | | | | |
| CoT-SC | 76.8 | 43.2 | 35.6 | 53.0 | 23.3 | 51.4 |
| LLM-J | 72.8 | 40.4 | 37.7 | 55.4 | 20.0 | 49.9 |
| ToT | 77.6 | 49.2 | 40.7 | 61.4 | 20.0 | 55.3 |
| ReST-MCTS | **81.4** | 50.7 | 40.3 | 62.6 | 20.2 | 56.9 |
| **APRM** (Ours) | 80.0 | **53.1** | **44.0** | **65.7** | **21.6** | **58.6** |

gold solution steps, the generator $G_\theta$ learns to perturb these solution steps. We alternate training between players, updating one for 5 gradient update steps while freezing the other. We use PPO (Schulman et al., 2017) with Optimistic Gradient Descent–Ascent (OGDA) (Mertikopoulos et al., 2019) as discussed in Section 2.4. $R_\phi$ is trained on a mixture of gold steps (50%) and negatives drawn from both current and past versions of $G_\theta$ (50%) to prevent forgetting. Hyperparameter details are given in the App L.

**Baselines** We compare against methods with comparable test time compute, which includes a) prompting-based methods b) trained reward models. Prompting based methods include CoT with self-consistency (k=5) (Wang et al., 2023), Tree of Thoughts (branching factor 5) (Yao et al., 2023) and LLM-as-a-Judge (Zheng et al., 2023) using same model as solver and as judge sampling a maximum of 5 times. Trained reward models include outcome-based approximations methods - AutoPSV (Lu et al., 2024a), trained

PRM using static datasets - Let's Verify Step by Step (Lightman et al., 2023c), MCTS based data generation techniques to train PRMs - Math-Shepherd (Lightman et al., 2023a), rStar-Math (Guan et al., 2025) and ReST-MCTS (Zhang et al., 2024), adversarially trained RMs (Bukharin et al., 2025), and PRMs trained by explicit hard sample mining. We re-implement all the methods using Llama-3.1-8B as a base model. Table 1 gives performance comparison with prompting and the best trained PRM baseline. A full comparison with all the baselines is given in Table 9.

For Hard-Neg. baseline, hard negatives are mined from static ReST-MCTS solver traces that assigns a value to each partial solution. To extract hard negatives, we select (a) step-level errors that do not lie on any correct trace but have high (b) "near-miss" nodes whose siblings lie on correct reasoning paths. We train on these mined negatives plus the corresponding positive steps. For the ARM (Bukharin et al., 2025) baseline, the paper traines an ORM, but for fair

*Table 2.* Performance comparison APRM with baselines on SciBench. Domains: atk=Atkins, cal=Calculus, cmc=ChemMC, cls=Classical, dif=Differential Eq., fun=Fundamentals, mat=Matter, qua=Quantum, sta=Statistics, the=Thermodynamics.

| Method | atk | cal | cmc | cls | dif | fun | mat | qua | sta | the | Overall |
|---|---|---|---|---|---|---|---|---|---|---|---|
| **GPT-OSS-120B** | | | | | | | | | | | |
| ToT | 58.3 | **96.2** | 43.5 | 51.6 | 74.1 | **37.5** | 46.4 | **50.0** | 89.1 | 61.0 | 61.4 |
| ReST-MCTS | 58.3 | 88.5 | 43.5 | 52.1 | 74.1 | 35.0 | 50.0 | **50.0** | 91.3 | 70.7 | 62.3 |
| **APRM** (Ours) | **60.3** | 91.2 | **44.8** | **52.4** | **76.9** | 36.6 | **51.5** | **50.0** | **92.5** | **73.1** | **64.0** |
| **GPT-OSS-20B** | | | | | | | | | | | |
| ToT | 60.0 | **88.5** | 43.5 | 45.2 | 74.1 | 37.5 | 46.4 | 60.0 | 89.1 | 65.9 | 61.7 |
| ReST-MCTS | 61.3 | 86.2 | 43.5 | 44.5 | 71.5 | 37.5 | 50.0 | 60.0 | 89.1 | **66.2** | 61.8 |
| **APRM** (Ours) | **61.8** | 86.1 | **44.8** | **47.6** | **75.1** | **38.8** | **52.0** | **61.3** | **91.0** | 65.3 | **63.0** |
| **Gemma-3-27B-IT** | | | | | | | | | | | |
| ToT | 60.0 | 92.3 | 43.5 | 48.4 | 77.8 | 30.0 | 50.0 | 50.0 | 89.1 | **63.4** | 61.1 |
| ReST-MCTS | 60.0 | 92.2 | 43.5 | 48.4 | 77.8 | 31.5 | 57.1 | 50.0 | 89.1 | 63.2 | 61.8 |
| **APRM** (Ours) | **62.2** | **94.6** | **45.1** | **50.2** | **80.2** | **31.9** | **59.3** | **52.0** | **91.4** | **63.4** | **63.5** |
| **Gemma-3-12B-IT** | | | | | | | | | | | |
| ToT | 43.3 | **73.1** | 34.8 | **51.6** | 59.3 | **40.0** | 21.4 | **20.0** | 80.4 | 36.6 | **47.6** |
| ReST-MCTS | 43.3 | 69.2 | 34.8 | 35.5 | 44.4 | 32.5 | 25.0 | 10.0 | 76.1 | 26.8 | 41.8 |
| **APRM** (Ours) | **44.8** | 71.5 | **35.9** | 36.8 | 45.8 | 33.6 | **25.9** | 10.4 | 78.4 | 27.8 | 43.2 |

comparison we trained a process reward model using their methodology. We train two PRMs using PRM800K (Lightman et al., 2023a), a human-labelled process level supervision dataset for math. We then train the generator to generate steps with low agreement between these PRMs, and trained a final PRM using this static generator.

**Test Setup** We evaluate on five math reasoning benchmarks: MATH500 (Hendrycks et al., 2021) (in-domain, in-distribution), JEEBench (Arora et al., 2023), Olympiad-Bench (He et al., 2024), AIME25 (Patel et al., 2024), and AMC (AI-MO, 2024) (in-domain, out-of-distribution). AIME25 contains recent examination questions, ensuring no data contamination. The effectiveness of trained PRMs is assessed by evaluating the performance of a solver model supervised by the PRM during test time. We sample a maximum of 5 times per step. We experiment on GPT and Gemma model families with different sizes (GPT-OSS-120B, GPT-OSS-20B, Gemma-3-12B-it and Gemma-3-27B-it) using API based inference. All experiments are run once with temperature 0. See App H for compute details.

### 3.2. Experimental Results

**APRM outperforms baselines** Table 1, shows the performance LLM solvers guided by our PRM against prior methods under equivalent test-time compute. Averaged over five benchmarks and four solver models, APRM improves solver accuracy by +3.4 pp relative to the strongest trained PRM baseline - ReST-MCTS (Zhang et al., 2024) and by +4.2 pp relative to the best prompting method. We observe the largest gains on JeeBench, which is an out-of-distribution benchmark, where APRM outperforms ReST-MCTS by +5.3

pp, suggesting better generalization beyond the training distribution.

**APRM remains effective as with solver scales** We evaluate whether APRM remains effective as the solver size increases. Table 1 shows that APRM's average improvement over prompt-based methods grows from +3.3 pp with Gemma-3-12B, to +4.3 pp with GPT-OSS-20B, and further to +5.0 pp with Gemma-3-27B. In contrast, PRMs trained on non-adaptive data (e.g. ReST-MCTS) show gains that are smaller and in some cases decline as solver size increases, dropping from +1.6 pp on Gemma-3-12B to −7.6 pp on Gemma-3-27B. These results support our hypothesis that adversarial training enables robust PRMs that remain effective even as solver models become stronger.

**APRM generalizes across domains** To test whether our adversarial training captures general error patterns rather than overfitting to math, we evaluate on SciBench (Wang et al., 2024b), comparing APRM with the best prompting and PRM baselines (Tab. 2). On GPT-OSS-20B, APRM reaches 63.0% accuracy versus 61.8% for ReST-MCTS and 61.1% for ToT. On GPT-OSS-120B, it achieves 64.0%, compared to 62.3% and 61.4%. This shows that APRM learns transferable properties of incorrect reasoning.

**APRM provides strong reward signal for RL Post-Training** To assess APRM's utility beyond inference-time search, we evaluate its effectiveness as a reward signal for RL post-training. We fine-tune Gemma-3-12B on the MATH dataset using GRPO, comparing APRM against standard outcome supervision and the strongest PRM baseline (ReST-MCTS). As shown in Table 3, APRM provides a significantly stronger learning signal, outperforming outcome-

*Table 3.* **RL Post-Training Performance.** Accuracy (%) of `Gemma-3-12B` fine-tuned using different reward signals. APRM serves as a more effective dense reward than baselines.

| Method | MATH500 | JEEB. | OlympiadB. | AIME25 | AMC |
|---|---|---|---|---|---|
| Base Model (CoT) | 76.8 | 43.2 | 35.6 | 53.0 | 23.3 |
| Final Correct (Outcome) | 80.0 | 48.2 | 40.0 | 59.8 | 21.0 |
| ReST-MCTS PRM | 83.0 | 51.6 | 43.2 | 61.4 | 23.3 |
| **APRM (Ours)** | **86.8** | **58.6** | **48.0** | **65.7** | **25.3** |

based RL by +6.8 pp on MATH500 and demonstrating superior generalization on OOD benchmarks like JEEBench (+10.4 pp). Training details are provided in Appendix L.

**Performance on PRM benchmarks**   We evaluate whether the predictions of `APRM` align with human annotations by comparing it's performance with the baselines on a subset of the PRM800K test set (Lightman et al., 2023b). As shown in Figure 4, we report accuracy, precision, recall, and F1 score for all methods. We find that `APRM` achieves the highest performance across all metrics, indicating stronger agreement with human-anotations compared to baselines.

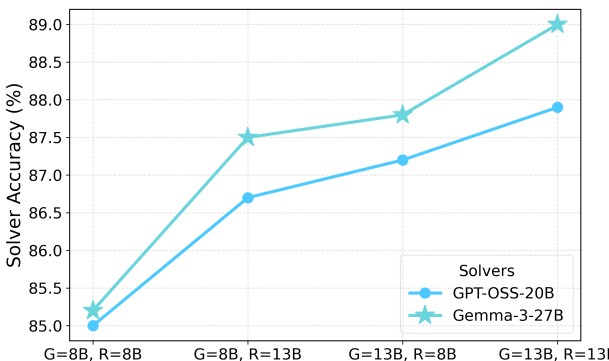

*Figure 2.* `APRM` augmented solver accuracy averaged across math reasoning benchmarks for different $R_\phi$ and $G_\theta$ backbones.

**Scaling properties of `APRM`**   We study how `APRM` scales with the backbone sizes of $G_\theta$ and $R_\phi$, using Llama-3.1-8B and Llama-2-13B in four settings: (1) $G$=8B, $R$=8B; (2) $G$=8B, $R$=13B; (3) $G$=13B, $R$=8B; and (4) $G$=13B, $R$=13B. Figure 2 reports solver accuracy averaged over all math benchmarks, showing monotonic gains from (1) to (4) with an overall improvement of $\sim$ 3.8 pp. We observe that gains are more when increasing the generator's capacity compared to increasing the reward model. We hypothesize that larger generators exposes the PRM to broader distribution reasoning errors.

### 3.3. Ablation Study

**Optimizer Ablation.**   We ablate on entropy regularization and the OGDA optimizer. Table 8 shows results on MATH, JEE, and OlympiadBench using `GPT-OSS-20B`.

*Table 4.* Oracle component ablation (`GPT-OSS-20B` solver accuracy, %). Each row removes one group of checks and retrains APRM. $\Delta$ is the change in average relative to the full oracle.

| Variant | MATH | JEE | Oly | Avg | $\Delta$ |
|---|---|---|---|---|---|
| APRM (full) | 91.0 | 73.0 | 90.4 | 84.8 | — |
| − Proximity | 88.4 | 70.2 | 87.6 | 82.1 | −2.7 |
| − Semantic | 87.2 | 69.0 | 86.4 | 80.9 | −3.9 |
| H1+H2 only | 88.0 | 70.0 | 87.8 | 81.9 | −2.9 |
| H3+H4+H5 | 89.6 | 71.4 | 88.8 | 83.3 | −1.5 |
| − Structural | 90.2 | 72.0 | 89.4 | 83.9 | −0.9 |
| H3 only, untrained | 80.4 | 52.6 | 66.2 | 66.4 | −18.4 |

Removing entropy regularization reduces accuracy by about 4 points, while removing OGDA has a similar effect. This complies with the theory, that each component helps stabilize training.

**Oracle ablation.**   We ablate each group of oracle checks by removing it and retraining APRM end to end (Table 4). The semantic validators (H1–H5) contribute most (−3.9 pp average), since they are what mathematically decide incorrectness. The proximity filters matter next (−2.7 pp): without them the generator produces degenerate, far-off perturbations that the reward model detects trivially. The comparison-based checks H3–H5 recover most of the signal, while the algebraic-legality check H3 alone, with no training, reaches only 66.4% average, far below every trained PRM.

**Oracle label quality.**   We compare the oracle against human judgment on 200 generator outputs labeled by two annotators (Cohen's $\kappa = 0.91$). The oracle reaches 93.9% precision and 86.0% recall overall. Precision stays above 91% even at training step $t = 3000$, while recall drops to 79.4% as the generator's errors become more subtle, which is exactly why the PRM must generalize beyond the oracle rather than memorize it.

**Frozen-generator ablation.**   To test whether co-evolution of the generator matters, we freeze the generator at different points and continue training only the PRM (Table 5). Freezing at initialization gives 64.5% average, barely above the base solver with no PRM (62.5%): an untrained generator rarely produces genuine errors, so the PRM sees few

*Table 5.* Frozen-generator ablation (`GPT-OSS-20B` solver accuracy, %). The generator is frozen at the given step and only the PRM continues training.

| Variant | MATH | JEE | Oly | Avg |
|---|---|---|---|---|
| APRM (full co-evolution) | 91.0 | 73.0 | 90.4 | 84.8 |
| Freeze G at $t = 2000$ | 89.8 | 71.4 | 88.6 | 83.3 |
| Freeze G at $t = 1000$ | 89.0 | 70.6 | 87.4 | 82.3 |
| Freeze G at init | 79.4 | 50.8 | 63.4 | 64.5 |
| Base solver (no PRM) | 78.0 | 48.4 | 61.0 | 62.5 |

*Table 7.* Generator error-type distribution (%) at two training stages, from 500 hand-annotated outputs.

| Error type | $t = 500$ | $t = 3000$ |
|---|---|---|
| Numeric substitution | 58 | 15 |
| Operator swap | 22 | 10 |
| Conceptual (wrong formula) | 8 | 32 |
| Logical (wrong conditional) | 5 | 25 |
| Structural (drop condition) | 4 | 11 |
| Entity swap | 3 | 7 |

negatives and learns to accept almost everything. The later the generator is frozen, the more, and harder, negatives the PRM receives, approaching full co-evolution (84.8%).

**Continuous reward.** The main results use a binary reward for the generator. We also test a continuous variant in which a successful deception is rewarded by the PRM's confidence that the incorrect step is correct, so a subtle error accepted with probability 0.95 gives reward 0.95 while a trivial error accepted with probability 0.55 gives only 0.55 (the $-1$ penalty is unchanged). The continuous reward gives a marginal gain ($+0.3$ pp average, Table 6), which we attribute to the adversarial setup already providing an implicit difficulty curriculum.

### 3.4. Qualitative Analysis

**Evolution of Generator perturbations.** We first analyze how the Generator's perturbations evolve during training. In the die-rolling problem (Fig. A), $G$ initially perturbs the step to make a shallow mistake of as miscounting the number of odd faces. By mid-training, it produces steps that look locally consistent but misinterpret the problem statement, e.g., doubling all faces without considering conditional rules. At later stages, it introduces subtle logical errors, like averaging probabilities instead of taking a weighted average. This trajectory suggests that adversarial interaction trains $G$ to generate progressively harder negatives for $R$.

**Generator error distribution.** To verify that the generator does not collapse to trivial edits, we annotate 500 generator outputs (two annotators, $\kappa = 0.87$) and track the error type across training (Table 7). Surface edits (numeric substitutions and operator swaps) fall from 80% at $t = 500$ to 25% at $t = 3000$, while conceptual and logical errors rise from 13% to 57%. The generator shifts from arithmetic

*Table 6.* Binary vs. continuous generator reward (`GPT-OSS-20B` solver accuracy, %).

| Reward | MATH | JEE | Oly | AIME | AMC | Avg |
|---|---|---|---|---|---|---|
| Binary | 91.0 | 73.0 | 90.4 | 93.6 | 68.0 | 85.0 |
| Continuous | 91.2 | 73.4 | 90.6 | 93.8 | 68.4 | 85.3 |

slips to genuine reasoning errors as training proceeds.

**APRM with different solver sizes.** We next examine `APRM`-trained PRMs when the solver capacity increases. In the shortest-path problem (Fig. B), GPT-oss-20B uses a non-coherent languages and produces a confusing and incorrect coordinate setup, while GPT-oss-120B generates a more coherent geometric reasoning but still mislocates the position of gecko. In both cases, `APRM` flags the faulty reasoning step, showing that adversarial training makes the PRM robust to errors even when the text is fluent.

**Cross-domain transfer.** We test whether `APRM` captures generalizable properties of flawed reasoning by applying it to a chemistry problem from SciBench (Fig. C). In this case, the solver neglected to convert centimeters to meters before applying $\Pi = \rho g h$. `APRM`-trained PRM flagged this unit error, which lead to correct final answer. This example suggests that adversarially generated negatives are not tied to math-specific syntax errors but can transfer to scientific reasoning tasks.

**Comparison with PRM baselines.** We further compare `APRM` with ReST-MCTS on a calculus reasoning problem (Fig. D). The problem asks to calculate the value of an expression at $x = 1$, where the numerator and denominator approach to zero. The solver factorizes and cancels the common factor and computes the value. This step appears valid, and even produces the correct final answer. However, this cancellation is invalid at $x = 1$ point, because it amounts to dividing by zero. ReST-MCTS fails to flag this subtle mistake, whereas, `APRM` flags this. This example highlights how adversarial training exposes PRMs to errors that lead to correct final answer, which static rollouts are not able to.

**Failure Cases.** Finally, we analyze the errors missed by `APRM`. We find that one recurring failure mode occurs when solvers invoke theorems without checking preconditions. For instance, in the series convergence problem (Fig. E), GPT-OSS-120B applied the Alternating Series Test to a sequence that does not alternate term-by-term, and the PRM accepted it. Similarly, in an inequality problem, the solver applied Jensen's inequality outside its domain of concavity,

and `APRM` again failed to flag the mistakes. We hypothesize that $G$ struggles to generate adversarial negatives that specifically target theorem preconditions, this might be a limitation due to the size of the backbone model used.

## 4. Related Work

**Mathematical Reasoning with LLMs**   LLM reasoning has advanced through chain-of-thought prompting (Wei et al., 2022), self-consistency (Wang et al., 2023), and outcome-level reinforcement learning (DeepSeek-AI, 2025; Jaech et al., 2024). Outcome-based reward models (Yu et al., 2024a) give sparse feedback (Cui et al., 2025) and can reinforce faulty reasoning. Process Reward Models (PRMs) (Luo et al., 2024; Cui et al., 2025; Havrilla et al., 2024; Lightman et al., 2023c; Lu et al., 2024a; Li & Li, 2025) address this by supervising intermediate steps, but require costly human annotations. Synthetic data generation methods (Lightman et al., 2023a; Zhang et al., 2024; Guan et al., 2025; Sun et al., 2025; Du et al., 2025; Li et al., 2024) generate synthetic step-level data, yet rely on final-answer correctness, which fails when wrong steps yield right answers. `APRM` instead trains a generator adversarially to produce hard, plausibly incorrect steps that target the PRM's weaknesses. This yields a dynamic curriculum of negatives, continuously adapting to the PRM's blind spots rather than relying on fixed heuristics.

**Adversarial Training and Robustness**   Adversarial training was introduced with GANs (Goodfellow et al., 2014), where a generator learns a target distribution byto fooling a discriminator, and later extended to robust optimization (Madry et al., 2018) and GAIL (Ho & Ermon, 2016) for policy imitation. `APRM` applies this idea to LLM reasoning steps, which are discrete and require a different setup. Unlike GANs or robust optimization that use zero-sum, single-objective formulations (Madry et al., 2018; Goodfellow et al., 2014), `APRM` defines a general-sum game with non-opposing utilities for the Generator and Reward Model. This yields equilibria that are stationary points of the game dynamics (Nash equilibria) rather than simple minimax solutions. `APRM` also connects to self-play in multi-agent RL (Baker et al., 2020) and recent advances in optimization for adversarial learning (Mertikopoulos et al., 2019;

Daskalakis et al., 2018). Further, more recently, Bukharin et al. (2025) applied adversarial training to outcome reward models using ensemble disagreement, and Zhang et al. (2025) use adversarially generated preferences to improve reward model generalization. In contrast, `APRM` targets process supervision with a step-level generator and a non-LLM oracle, and evolves the generator and PRM in tandem rather than using a static generator.

## 5. Conclusion, Limitations and Future Work

In this work, we introduced `APRM`, an adversarial framework to improve process supervision for LLM reasoning. Unlike prior approaches that rely on outcome-based feedback, static human-annotated datasets, or heuristic search with MCTS, `APRM` actively optimizes for hard negatives to train the PRM. This adversarial setup exposes the PRM to subtle reasoning errors making it robust and generalizable across domains.

A limitation of our method is the computational cost. While inference-time compute is unchanged, training requires additional resources due to the adversarial optimization between generator and PRM. This cost is incurred only once, but remains higher than approaches based solely on static datasets.

As the availability of high-quality training data for large models plateaus, adversarial training offers a way to generate targeted supervision, specifically focusing on model weaknesses, where data collection is expensive or impractical. This makes the approach relevant beyond mathematical reasoning, with potential applications in scientific domains, robotics, and other areas where annotation is difficult to scale.

Future work includes extending the framework beyond the two-player formulation. Multi-agent variants introduce diverse adversarial roles that target different error modes. Another direction is applying the approach to embodied agents and robotics, where agents must learn to operate in noisy environments induced by other agent's adversarial interactions. A major direction to explore is how to generalize this to a general multi agent adversarial setting to learn on general tasks. On the optimization side, although our analysis established convergence under OGDA with regularization, further work is needed to understand large-scale training dynamics and to develop algorithms that enhance stability.

## Impact Statement

This paper presents work whose goal is to advance the field of Machine Learning. There are many potential societal consequences of our work, none which we feel must be specifically highlighted here.

*Table 8.* Ablation study on `GPT-OSS-20B`. Accuracy (%) on representative benchmarks.

| Variant | MATH | JEE | Oly | Overall |
|---|---|---|---|---|
| `APRM` (full) | **91.0** | **73.0** | **90.0** | **85.2** |
| – no entropy reg. | 86.4 | 69.4 | 85.5 | 80.9 |
| – no OGDA | 84.6 | 67.9 | 83.7 | 79.2 |
| – both removed | 83.6 | 67.2 | 82.8 | 78.4 |

**Use of Large Language Models.** In compliance with the ICML guidelines, we used LLMs only to make the authors' own writing more concise and grammatically clear; all ideas, experiments, equations, and results are the authors' own. Appendix M gives before-and-after examples.

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

## A. Evolution of Generator Perturbations

**Question**: A die, with the numbers 1, 2, 3, 4, 6, and 8 on its six faces, is rolled. After this roll, if an odd number appears on the top face, all odd numbers on the die are doubled. If an even number appears on the top face, all the even numbers are halved. If the given die changes in this way, what is the probability that a 2 will appear on the second roll of the die?

**Correct Step**: The die has six faces. There are two odd numbers 1, 3 and four even numbers 2, 4, 6, 8. Therefore, the probability of rolling an odd number is 2/6 = 1/3, and the probability of rolling an even number is 4/6 = 2/3.

- - - - - - - - - - - - - - - - - - - - - - - - - - - - - - - - - - - - - - - - - - - - - - - - - - - - - - - - - -

**t = 500**: The die has six faces. There are three odd numbers 1, 3 and three even numbers 2, 4, 6, 8. Therefore, the probability of rolling an odd number is 3/6 = 1/2 , and the probability of rolling an even number is 3/6 = 1/2 .

**Comment**: Trivial mistake of counting number of odds wrong.

- - - - - - - - - - - - - - - - - - - - - - - - - - - - - - - - - - - - - - - - - - - - - - - - - - - - - - - - - -

**t = 1000**: We first determine the two possible states of the die for the second roll. If an odd number (1 or 3) is rolled first, all numbers on the die are doubled, resulting in a new die with faces 2, 4, 6, 8, 12, 16.

**Comment**: wrongly interpreted question, seemingly correct step on it's own but wrong in context of the problem.

- - - - - - - - - - - - - - - - - - - - - - - - - - - - - - - - - - - - - - - - - - - - - - - - - - - - - - - - - -

**t = 3000**: We must find the probability of rolling a 2. There are two scenarios. First roll is odd (P=1/3): The die becomes 2, 2, 6, 4, 6, 8. The probability of rolling a 2 here is 2/6. First roll is even (P=2/3): The die becomes 1, 1, 3, 2, 3, 4. The probability of rolling a 2 here is 1/6. To find the total probability, we average these two probabilities: (2/6 + 1/6) / 2 = (3/6) / 2 = 1/4.

**Comment**: This error is deceptive because every calculation is correct, there is a conceptual error, probabilities are not averaged like this. Weighted average should be used here.

## B. PRM across solver model sizes

**Question**: A gecko is in a room that is 12 feet long, 10 feet wide and 8 feet tall. The gecko is currently on a side wall ($10'$ by $8'$), one foot from the ceiling and one foot from the back wall ($12'$ by $8'$). The gecko spots a fly on the opposite side wall, one foot from the floor and one foot from the front wall. What is the length of the shortest path the gecko can take to reach the fly assuming that it does not jump and can only walk across the ceiling and the walls? Express your answer in simplest radical form.

**GPT-OSS-20B**

We need shortest path across surfaces of a rectangular box. Geometric unfolding problem. Gecko on side wall: side wall dimensions 10' by 8'. Which side? They say side wall (10' by 8') one foot from ceiling and one foot from back wall (12' by 8'). So coordinate system: room dimensions: length 12 (x), width 10 (y), height 8 (z). Let's set x along length (front to back), y along width (left to right), z vertical. Back wall: plane x=12? Actually back wall is 12' by 8' (length x height). So back wall is at x=12, facing front. Good.

- - - - - - - - - - - - - - - - - - - - - - - - - - - - - - - - - - - - - - - - - -

**GPT-OSS-120B**

Step 1 – Set up a coordinate system
Let

- the $x$-axis run from the front wall $(x = 0)$ to the back wall $(x = 12)$;

- the $y$-axis run from the left side wall $(y = 0)$ to the right side wall $(y = 10)$;

- the $z$-axis run from the floor $(z = 0)$ to the ceiling $(z = 8)$.

The gecko starts on the left side wall $(y = 0)$ one foot below the ceiling and one foot in front of the back wall:S=(x,y,z)=(11,0,7).

## C. Comparision to ReST-MCTS

**Question:** Evaluate
$$F(x) = \frac{x^5 - 1}{x - 1}$$
at $x = 1$.

Factor the numerator: $\frac{x^5-1}{x-1} = \frac{(x-1)(x^4+x^3+x^2+x+1)}{x-1} = x^4 + x^3 + x^2 + x + 1.$ Now substitute $x = 1$:

$$F(1) = 1^4 + 1^3 + 1^2 + 1 + 1 = 5.$$

**Comment:** The cancellation step is invalid at $x = 1$, because it amounts to dividing by zero. ReST-MCTS approves the solution but `APRM` Flags the error.

## D. Domain Transfer

**Question**: The osmotic pressures of solutions of poly(vinyl chloride) (PVC) in cyclohexanone at 298 K are given below. The pressures are expressed in terms of the heights of solution (of mass density $\rho = 0.980$ g cm$^{-3}$) in balance with the osmotic pressure. Determine the molar mass of the polymer.

| $c$ (g dm$^{-3}$) | 1.00 | 2.00 | 4.00 | 7.00 | 9.00 |
|---|---|---|---|---|---|
| $h$ (cm) | 0.28 | 0.71 | 2.01 | 5.10 | 8.00 |

Step 1 – Relate the measured height to the osmotic pressure
The height $h$ of the liquid column balances the osmotic pressure $\Pi$:

$$\Pi = \rho g h$$

$$\rho = 0.980 \text{ g cm}^{-3} = 980 \text{ kg m}^{-3}, \qquad g = 9.81 \text{ m s}^{-2}$$

$$\rho g = 980 \times 9.81 \approx 9.61 \times 10^3 \text{ Pa m}^{-1}$$

$$\Pi \approx 9.61 \times 10^3 \times h \text{ Pa}$$

| $h$ (cm) | $\Pi$ (Pa) |
|---|---|
| 0.28 | 26.9 |
| 0.71 | 68.2 |
| 2.01 | 192.0 |
| 5.10 | 489.8 |
| 8.00 | 768.3 |

## E. Failure Cases

**Question**: Determine the convergence of

$$\sum_{n=1}^{\infty} \frac{(-1)^{\lfloor \sqrt{n} \rfloor}}{\sqrt{n}}.$$

Let $a_n = \frac{(-1)^{\lfloor \sqrt{n} \rfloor}}{\sqrt{n}}$.

Since $1/\sqrt{n} \to 0$ and the terms alternate in sign, the Alternating Series Test implies convergence. Moreover, grouping terms by blocks where $\lfloor \sqrt{n} \rfloor$ is constant yields partial cancellations, so $\sum_{n=1}^{\infty} |a_n| = \sum_{n=1}^{\infty} \frac{1}{\sqrt{n}}$ is effectively controlled by these cancellations, implying absolute convergence.

**Comment** The sequence does not alternate term-by-term. The convergence is conditional.

**Question**: For $a, b, c > 0$ with $a + b + c = 3$, decide whether

$$\frac{a}{1+a^2} + \frac{b}{1+b^2} + \frac{c}{1+c^2} \leq \frac{3}{2}$$

holds.

Let $f(x) = \frac{x}{1+x^2}$. Since $f(x)$ is concave, by Jensen's inequality,

$$\frac{a}{1+a^2} + \frac{b}{1+b^2} + \frac{c}{1+c^2} = \sum f(a) \leq 3f\left(\frac{a+b+c}{3}\right) = 3f(1) = \frac{3}{2}.$$

**Comment** $f(x)$ is concave only on the interval $0, \sqrt{3}$. Jensen's application is not guaranteed outside this interval and the proof is invalid even though the conclusion correct.

## F. Correctness Oracle

Given the gold step $s_i$ and a candidate step $s_i'$ produced by the generator, the oracle computes a set of proximity filters (ensure $s_i'$ is close to $s_i$ but not a trivial copy), hard semantic validators (mathematically disprove/verify correctness), structural validators (check symbol/number consistency), and optional execution checks. The oracle outputs $y(s_i') \in \{0, 1\}$ for INCORRECT or CORRECT

**Proximity filters (all must pass).** These constrain the candidate to be a plausible variant of $s_i$ (prevents degenerate far-off edits and trivial copies). Let $\text{len}(\cdot)$ be token length, $\cos(\cdot, \cdot)$ a sentence embedding cosine similarity, and $\text{Lev}(\cdot, \cdot)$ a normalized Levenshtein distance in $[0, 1]$.

$$
\begin{aligned}
&\textbf{(P1) Length window:} && \mathbb{W}_{\text{len}} = \mathbb{I}\left[\tfrac{1}{2}\text{len}(s_i) \leq \text{len}(s_i') \leq 2\,\text{len}(s_i)\right] \\
&\textbf{(P2) Similarity band:} && \mathbb{W}_{\text{sim}} = \mathbb{I}\left[\tau_{\min} \leq \cos(s_i', s_i) \leq \tau_{\max}\right], \quad (\tau_{\min}{=}0.5,\ \tau_{\max}{=}0.9) \\
&\textbf{(P3) Non-exactness:} && \mathbb{W}_{\text{neq}} = \mathbb{I}\left[s_i' \neq s_i\right] \\
&\textbf{(P4) Minimality:} && \mathbb{W}_{\text{min}} = \mathbb{I}\left[\text{Lev}(s_i', s_i) \leq \delta_{\max}\right], \quad (\delta_{\max}{=}0.35)
\end{aligned}
$$

We require $\mathbb{W}_{\text{prox}} \triangleq \mathbb{W}_{\text{len}} \cdot \mathbb{W}_{\text{sim}} \cdot \mathbb{W}_{\text{neq}} \cdot \mathbb{W}_{\text{min}} = 1$. If the proximity filters pass, we move on to check for the semantic validators.

**Hard semantic validators (decide correctness/incorrectness).** We compile both $s_i$ and $s_i'$ to symbolic forms when applicable (algebraic expressions, equalities, transforms). Let $\text{SymEq}(\cdot)$ denote a symbolic expression extracted from a step (when unavailable, the check abstains).

$$
\textbf{(H1) Symbolic equivalence:} \quad \mathbb{W}_{\text{eq}} = \begin{cases} 0 & \text{if } \text{simplify}(\text{SymEq}(s_i') - \text{SymEq}(s_i)) = 0 \\ \perp & \text{if symbolic forms unavailable} \\ 1 & \text{otherwise} \end{cases}
$$

$$
\textbf{(H2) Numeric consistency:} \quad \mathbb{W}_{\text{num}} = \begin{cases} 0 & \text{if } \max_{x \in \mathcal{T}} \left|\text{eval}_x(s_i') - \text{eval}_x(s_i)\right| \leq \epsilon \\ 1 & \text{if } \exists x \in \mathcal{T} \text{ s.t. diff} > \epsilon \\ \perp & \text{if not evaluable} \end{cases}
$$

**(H3) Algebraic legality:** $\quad \mathbb{W}_{\text{alg}} = \mathbb{I}\left[\text{illegal ops (e.g., divide-by-zero at the evaluation point)}\right]$

**(H4) Theorem preconditions:** $\quad \mathbb{W}_{\text{pre}} = \mathbb{I}\left[\text{some invoked preconditions don't hold}\right]$

**(H5) Units/dimensions:** $\quad \mathbb{W}_{\text{unit}} = \mathbb{I}\left[\text{dimensionally inconsistent; implicit unit swaps}\right]$

**Structural validators (local consistency).** Let $\text{Ent}(\cdot)$ be the multiset of named entities, numbers, and operators; $\text{Bind}(\cdot)$

maps symbols to their roles.

**(S1) Entity/number alignment:** $\quad \nVdash_{\mathrm{ent}} = \mathbb{I}\big[\mathrm{Jaccard}\big(\mathrm{Ent}(s_i'), \mathrm{Ent}(s_i)\big) \in [\alpha_{\min}, \alpha_{\max}]\big]$

**(S2) Variable binding consistency:** $\quad \nVdash_{\mathrm{bind}} = \mathbb{I}\big[\mathrm{Bind}(s_i') \text{ is not consistent with } \mathrm{Bind}(s_i)\big]$

We use $(\alpha_{\min}, \alpha_{\max}) = (0.5, 0.95)$ to allow small edits but prevent verbatim copies or unrelated content.

We flag a step as incorrect if any hard or structural validator fires, i.e. $\nVdash_{\mathrm{eq}} \vee \nVdash_{\mathrm{num}} \vee \nVdash_{\mathrm{alg}} \vee \nVdash_{\mathrm{pre}} \vee \nVdash_{\mathrm{unit}} \vee \nVdash_{\mathrm{ent}} \vee \nVdash_{\mathrm{bind}} = 1$.

**Thresholds.** We set $(\tau_{\min}, \tau_{\max}, \delta_{\max}, \alpha_{\min}, \alpha_{\max}, \epsilon) = (0.5, 0.9, 0.35, 0.5, 0.95, 10^{-6})$ by validation. Results are robust in a neighborhood of these values.

### F.1. Implementation Details

**Symbolic parsing.** We extract mathematical expressions from each natural-language step with a regular-expression parser that matches patterns such as "variable = expression", "LHS = RHS", and (in)equalities in LaTeX. For the step "so $6w + 6 = 54$, giving $6w = 48$ and $w = 8$", the parser returns the three expressions $\{6w + 6 = 54, \ 6w = 48, \ w = 8\}$, which are converted to `SymPy` objects; $\mathsf{SymEq}(s)$ denotes the set of expressions extracted from step $s$. When a step has no parseable mathematics (for example "we consider two cases"), SymEq returns $\perp$ and H1 abstains.

**Symbolic and numeric checks.** For H1 we test whether $\mathsf{simplify}\big(\mathsf{SymEq}(s_i') - \mathsf{SymEq}(s_i)\big) = 0$ using `sympy.simplify`, which handles algebraic identities such as $(x + 1)^2$ vs. $x^2 + 2x + 1$. H2 is a numeric fallback used when symbolic simplification is inconclusive: we evaluate both expressions at five random assignments per free variable and flag a mismatch if any pair differs by more than $\epsilon = 10^{-6}$.

**Theorem preconditions (H4).** H4 checks whether a perturbation changes which theorem preconditions are invoked. We keep a lookup table of roughly fifteen common theorems and their preconditions (for example, Jensen's inequality requires convexity on the interval, L'Hôpital's rule requires a $0/0$ or $\infty/\infty$ form, and the alternating series test requires alternating signs with monotone decrease). Theorem names are detected by keyword matching, and H4 fires if the perturbed step violates a precondition that the gold step satisfied. The full table is released with the code.

**Checks we considered and dropped.** We dropped two checks. Full dimensional-analysis tracking on intermediate expressions had a high false-positive rate when extracting physical dimensions from free-form text. A logical-connective check on "if-then" and "therefore" chains was unreliable with a regex parser, and we did not want the oracle to depend on an LLM. The released code includes the parser grammar, the theorem table, and the full validator implementation.

## G. KL and Entropy Regularization Prevent Mode Collapse

We provide formal results showing that KL- and entropy-based regularization guarantees full-support equilibria and thus prevents mode collapse.

**Lemma G.1** (Equivalence of KL Penalty and Entropy Bonus). *Let $\pi, \pi_0$ be distributions over a finite action space $\mathcal{A}$. Then*

$$-\mathrm{KL}(\pi \,\|\, \pi_0) = H(\pi) + \sum_{a \in \mathcal{A}} \pi(a) \log \pi_0(a),$$

*where $H(\pi) = -\sum_{a \in \mathcal{A}} \pi(a) \log \pi(a)$ is the Shannon entropy.*

**Lemma G.2** (Form of the Regularized Optimum). *Consider the objective*

$$J(\pi) = \sum_{a \in \mathcal{A}} \pi(a) V(a) \ - \ \tau \, \mathrm{KL}(\pi \,\|\, \pi_0),$$

*where $V(a)$ is the expected payoff or reward of action $a$. Then the maximizer is*

$$\pi^*(a) = \frac{1}{Z} \, \pi_0(a) \, \exp\!\Big(\tfrac{1}{\tau} V(a)\Big),$$

*with normalization constant $Z = \sum_a \pi_0(a) \exp(V(a)/\tau)$.*

*Proof.* By Lemma 1,

$$J(\pi) = \sum_a \pi(a)V(a) + \tau H(\pi) + \tau \sum_a \pi(a) \log \pi_0(a).$$

This is strictly concave in $\pi$. Introducing a Lagrangian with multiplier $\lambda$ for the constraint $\sum_a \pi(a) = 1$:

$$\mathcal{L}(\pi, \lambda) = \sum_a \pi(a)V(a) + \tau H(\pi) + \tau \sum_a \pi(a) \log \pi_0(a) + \lambda\Big(1 - \sum_a \pi(a)\Big).$$

Taking derivatives:

$$\frac{\partial \mathcal{L}}{\partial \pi(a)} = V(a) - \tau \log \pi(a) + \tau \log \pi_0(a) - \lambda.$$

Setting to zero and solving for $\pi(a)$ yields

$$\pi(a) \propto \pi_0(a) \exp\big(V(a)/\tau\big).$$

Normalization gives the Gibbs–Boltzmann form. $\square$

**Definition G.3** (Support of a Probability Distribution)**.** Given a probability distribution $P$ over a set $\Omega$, the support of $P$, denoted $\mathrm{supp}(P)$, is the set of all elements in $\Omega$ that are assigned a non-zero probability by $P$:

$$\mathrm{supp}(P) = \{\omega \in \Omega \mid P(\omega) > 0\}.$$

Now, we combine these to define full support for a policy, specifically in the context of an equilibrium:

**Definition G.4** (Full Support of an Equilibrium Policy)**.** An equilibrium policy $\pi_i^*$ for player $i$ (part of a Nash Equilibrium $\pi^* = (\pi_1^*, \ldots, \pi_N^*)$) is said to have **full support** if every pure strategy available to player $i$ has a non-zero probability of being played under that policy. Formally, for all $s \in S_i$:

$$\pi_i^*(s) > 0.$$

Equivalently, the support of the equilibrium policy is the entire pure strategy set: $\mathrm{supp}(\pi_i^*) = S_i$.

In simpler terms, if an equilibrium policy has full support, it means the player randomizes over all their available actions, never completely ruling out any single pure strategy.

**Theorem G.5** (Full Support of the Equilibrium Policy)**.** *If $\pi_0$ has full support over $\mathcal{A}$, then the solution $\pi^*$ from Lemma 2 also has full support.*

*Proof.* For each $a \in \mathcal{A}$,

$$\pi^*(a) = \tfrac{1}{Z}\,\pi_0(a)\exp(V(a)/\tau).$$

Since $\pi_0(a) > 0$ by assumption and the exponential is strictly positive, $\pi^*(a) > 0$. $\square$

**Corollary G.6** (Avoidance of Mode Collapse)**.** *Under the assumption of a full-support reference policy $\pi_0$, the generator's KL-regularized optimization problem admits a unique maximizer $\pi^*$ that assigns nonzero probability to every action. Hence the generator cannot collapse to a deterministic or low-support distribution.*

**Remark (Role of Temperature $\tau$).** The parameter $\tau$ controls exploration:

- As $\tau \to 0$, $\pi^*$ concentrates near maximizers of $V(a)$ (greedy strategy).

- As $\tau \to \infty$, $\pi^*$ approaches $\pi_0$.

Thus $\tau$ interpolates between exploitation and exploration while preserving full support.

## H. Compute-Performance Tradeoff

The compute time (GPU hours) required to train ReST-MCTS for 3 epochs (as recommended in the paper) and `APRM` for 5000 time steps (1 epoch). As we can see in the Figure 3, for a 0.25x increase in training time, `APRM` gives a 4.6% performance improvement.

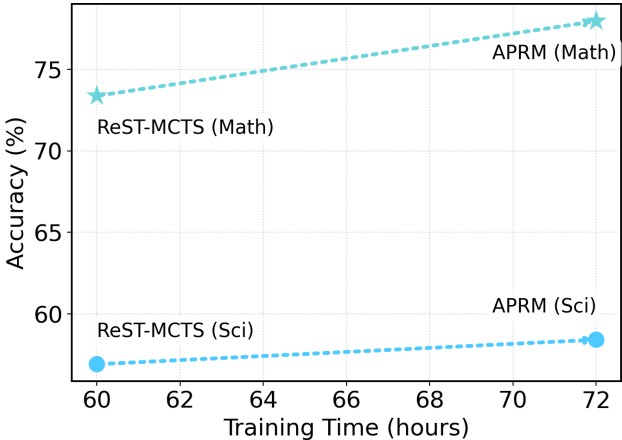

*Figure 3.* Training compute vs. accuracy on Math and SciBench. Dotted arrows connect ReST-MCTS (60 h) to `APRM` (72 h), showing the direction of improvement. `APRM` achieves higher accuracy on both benchmarks at slightly higher compute.

## I. Proof of Theorem 2

### Proof of Linear Time Convergence for OGDA with Strongly Monotone Operator

In our game, players' strategies are $\pi = (\pi_\theta, \pi_\phi) \in \diamond$, and their coupled dynamics are governed by a game operator $F : \Pi \to \Pi$. A Nash Equilibrium $z^* = (\theta^*, \phi^*)$ is a fixed point such that $F(z^*) = 0$.

**Definition I.1** (Strong Monotonicity). An operator $F : \Pi \to \Pi$ is $\mu$-strongly monotone if there exists $\mu > 0$ such that for all $x, y \in \Pi$:

$$\langle F(x) - F(y), x - y \rangle \geq \mu \|x - y\|^2.$$

**Definition I.2** (Lipschitz Continuity). An operator $F(is)L$-Lipschitz continuous if there exists $L > 0$ such that for all $x, y \in \Pi$:

$$\|F(x) - F(y)\| \leq L \|x - y\|.$$

Given a strongly monotone operator, there exists a unique $z^*$ such that $F(z^*) = 0$. The Optimistic Gradient Descent Ascent (OGDA) update rule is given by:

$$z^{(t+1)} = z^{(t)} - \eta \left( 2F(z^{(t)}) - F(z^{(t-1)}) \right),$$

where $\eta > 0$ is the learning rate, and for $t = 0$, we set $z^{(-1)} = z^{(0)}$.

**Theorem I.3** (Linear Convergence of OGDA). *Let $F(z)$ be an operator that is $\mu$-strongly monotone and $L$-Lipschitz continuous. Then, for a sufficiently small learning rate $\eta > 0$ (specifically, $\eta \leq \mu L^2$), the OGDA algorithm converges linearly to the unique Nash Equilibrium $z^*$ satisfying $F(z^*) = 0$. That is, there exist constants $C > 0$ and $\rho \in (0, 1)$ such that the distance to the equilibrium satisfies:*

$$\|z^{(t)} - z^*\|^2 \leq C\rho^t.$$

*Proof.* Let $e^{(t)} = z^{(t)} - z^*$ denote the error at iteration $t$. Since $F(z^*) = 0$, we can rewrite the OGDA update rule in terms of the error:

$$e^{(t+1)} = e^{(t)} - \eta \left( 2(F(z^{(t)}) - F(z^*)) - (F(z^{(t-1)}) - F(z^*)) \right).$$

Let $G^{(t)} = F(z^{(t)}) - F(z^*)$. Then,

$$e^{(t+1)} = e^{(t)} + \eta(2G^{(t)} - G^{(t-1)}).$$

Consider a quadratic Lyapunov function $V_t = \|e^{(t)}\|^2 + \alpha \|e^{(t)} - e^{(t-1)}\|^2$ for some $\alpha > 0$ to be chosen. For simplicity and conciseness, we present a direct argument for contraction.

By analyzing the dynamics of the error $e^{(t)}$, one can establish a contraction property. The proof typically involves studying the evolution of a combined state vector, e.g., $[e^{(t)}; e^{(t-1)}]$, or by carefully choosing a specific quadratic potential function.

Following established results for OGDA on strongly monotone operators, we know that for a sufficiently small learning rate $\eta$, the algorithm exhibits a contraction.

Specifically, for $\eta$ chosen such that $\eta^2 L^2 < \mu\eta$, and with additional conditions on $\eta$ to stabilize the "optimistic" step, it can be shown that there exist constants $\alpha_1, \alpha_2 > 0$ and $\rho \in (0, 1)$ such that for a suitably defined potential function (e.g., $W_t = \|e^{(t)}\|^2 + \alpha_1\langle e^{(t)}, e^{(t-1)}\rangle + \alpha_2\|e^{(t-1)}\|^2$):

$$W_{t+1} \leq \rho W_t.$$

This contraction arises from leveraging both the strong monotonicity and Lipschitz continuity properties. The strong monotonicity term $\langle F(z^{(t)}) - F(z^*), z^{(t)} - z^*\rangle \geq \mu\|z^{(t)} - z^*\|^2$ provides a robust "pull" towards the equilibrium. The Lipschitz continuity bounds the change in the gradient, controlling the step size. The "optimistic" term $F(z^{(t-1)})$ helps in dampening oscillations inherent in standard gradient ascent/descent for games.

For $\eta \in (0, \mu L^2)$, a more precise analysis demonstrates that the sequence of iterates $\{z^{(t)}\}$ converges linearly to $z^*$. The specific choice of $\eta$ balances the strong monotonicity and Lipschitz constant. For instance, in some analyses, choosing $\eta < 1L$ and further restricting it based on $\mu$ ensures contraction.

The linear convergence, also known as exponential convergence, implies that the error shrinks by a constant factor $\rho < 1$ at each iteration, leading to the bound $\|z^{(t)} - z^*\|^2 \leq C\rho^t$. □

## J. Performance on MATH

*Table 9.* Performance comparison of `APRM` with the prior methods on math reasoning benchmarks.

| Method | MATH500 | JEEB. | OlympiadB. | AIME25 | AMC | Avg. |
|---|---|---|---|---|---|---|
| **GPT-OSS-120B** | | | | | | |
| CoT-SC | 84.4 | 62.8 | 83.9 | 85.5 | 60.0 | 77.2 |
| LLM-J | 85.2 | 66.0 | 83.1 | 90.4 | **80.0** | 79.3 |
| ToT | 87.6 | 66.4 | 84.3 | 89.2 | 66.7 | 79.9 |
| AutoPSV | 86.0 | 65.0 | 84.0 | 88.0 | 65.0 | 78.5 |
| Verify SbS. | 85.0 | 63.5 | 83.5 | 86.0 | 61.0 | 77.8 |
| Math S. | 85.8 | 64.5 | 83.8 | 87.5 | 62.0 | 78.3 |
| rStar Math | 90.0 | 68.0 | 84.3 | 90.7 | 67.0 | 80.8 |
| ReST-MCTS | **91.6** | 69.1 | 84.3 | 91.0 | 67.4 | 81.7 |
| **APRM** (Ours) | 91.4 | **70.3** | **89.4** | **94.5** | 70.7 | **83.0** |
| **GPT-OSS-20B** | | | | | | |
| CoT-SC | 78.0 | 48.4 | 61.0 | 69.9 | 33.3 | 62.2 |
| LLM-J | 85.2 | 68.0 | 83.9 | 92.8 | 60.0 | 80.3 |
| ToT | 87.6 | 67.6 | 85.6 | 90.4 | 66.7 | 80.7 |
| AutoPSV | 84.0 | 60.0 | 75.0 | 85.0 | 50.0 | 70.0 |
| Verify SbS. | 80.0 | 55.0 | 70.0 | 80.0 | 45.0 | 68.0 |
| Math S. | 82.0 | 58.0 | 73.0 | 82.0 | 48.0 | 71.5 |
| rStar Math | 89.0 | 68.2 | 84.3 | 90.8 | 65.5 | 81.5 |
| ReST-MCTS | **91.0** | 68.3 | 84.7 | 91.3 | 66.0 | 82.3 |
| **APRM** (Ours) | **91.0** | **73.0** | **90.4** | 93.6 | **68.0** | **85.0** |
| **Gemma-3-27B** | | | | | | |
| CoT-SC | 78.8 | 48.8 | 48.7 | 69.9 | 13.3 | 58.4 |
| LLM-J | 86.4 | 69.2 | 83.1 | 91.6 | 73.3 | 80.4 |
| ToT | 88.0 | 67.6 | 85.2 | 88.0 | 60.0 | 80.2 |
| AutoPSV | 83.0 | 60.0 | 70.0 | 80.0 | 40.0 | 65.0 |
| Verify SbS. | 80.0 | 55.0 | 60.0 | 75.0 | 30.0 | 63.0 |
| Math S. | 81.5 | 58.0 | 65.0 | 78.0 | 35.0 | 66.0 |
| rStar Math | 89.5 | 69.8 | 83.8 | 89.5 | 60.5 | 72.0 |
| ReST-MCTS | **93.2** | 70.3 | 84.3 | 90.6 | **61.2** | 72.6 |
| **APRM** (Ours) | 91.4 | **74.4** | **90.7** | 91.9 | 60.8 | **85.2** |
| **Gemma-3-12B** | | | | | | |
| CoT-SC | 76.8 | 43.2 | 35.6 | 53.0 | 23.3 | 51.4 |
| LLM-J | 72.8 | 40.4 | 37.7 | 55.4 | 20.0 | 49.9 |
| ToT | 77.6 | 49.2 | 40.7 | 61.4 | 20.0 | 55.3 |
| AutoPSV | 78.0 | 45.0 | 38.0 | 58.0 | 20.1 | 54.0 |
| Verify SbS. | 77.5 | 42.0 | 36.0 | 55.0 | 20.05 | 52.0 |
| Math S. | 78.5 | 44.0 | 37.0 | 57.0 | 20.1 | 53.0 |
| rStar Math | 79.0 | 50.0 | 39.0 | 62.0 | 20.1 | 56.0 |
| ReST-MCTS | **81.4** | 50.7 | 40.3 | 62.6 | 20.2 | 56.9 |
| **APRM** (Ours) | 80.0 | **53.1** | **44.0** | **65.7** | **21.6** | **58.6** |

# K. Performance on PRM Benchmarks

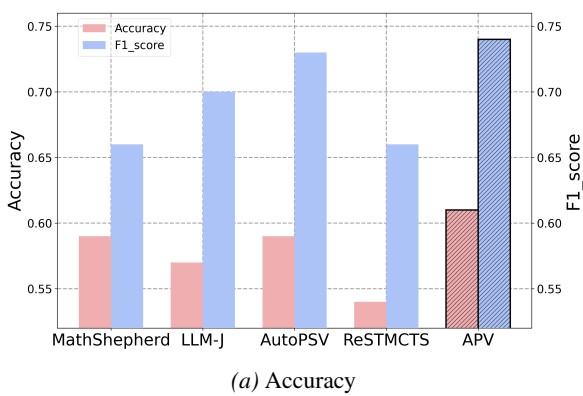 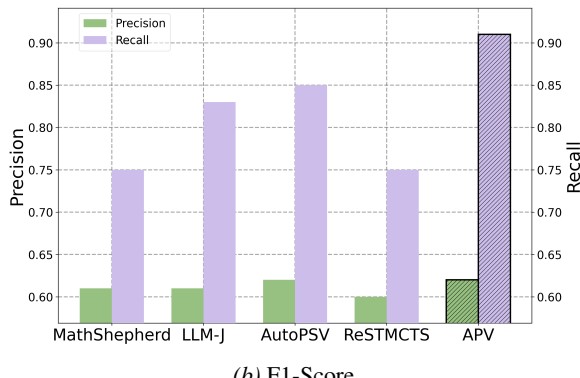

*(a)* Accuracy          *(b)* F1-Score

*Figure 4.* Comparison of performance on a subset of PRM800K (Lightman et al., 2023b). `APRM` (hatched) achieves the highest accuracy, precision, recall, and F1 score, indicating stronger alignment with humans

# L. Training Details

## L.1. Hyperparameters to train `APRM`

Embedding Model (for correctness oracle): sentence-transformers/all-MiniLM-L6-v2
PPO Hyperparameters (Reinforcement Learning):
Learning Rate $\eta$: 1e-6
Optimizer: OGDA
PPO Batch Size : 4
PPO Epochs (mini epoch'): 2
$\tau$ :0.01
$c_H$: 0.01
Clipping Range ($\epsilon$ for PPO loss):0.2
Discount Factor ($\gamma$): 0.99
Generalized Advantage Estimation (GAE) Lambda ($\lambda$):0.95
Maximum New Tokens: 256; Mixed Precision used bf16 managed by accelerate.

## L.2. Additional Implementation Details

### 1. ARM Baseline Implementation

- **Backbone:** We used `Llama-3.1-8B-Instruct` as the backbone for both the Process Reward Models (PRMs) and the Generator.

- **Outcome RM Training:** We trained two separate PRMs on the PRM800K (Lightman et al., 2023b) dataset using the Bradley-Terry objective with different random seeds. *Note: While the original ARM paper (Bukharin et al., 2025) trains Outcome Reward Models, we train Process Reward Models to suit our task setup, preventing the use of the exact original training data.*

  - **Optimization:** AdamW optimizer, Learning Rate $1 \times 10^{-6}$, Batch Size 64, trained for 1 Epoch.

- **Generator Training:** We employed RLOO (Reinforce Leave-One-Out) to train the generator, as specified in the original paper.

  - **Optimization:** AdamW optimizer, Learning Rate $5 \times 10^{-7}$, Batch Size 64.
  - **Sampling:** We generated 16 samples per prompt during training.

- **Final PRM Training Data Selection:** To construct the training data for the final model, we generated 16 samples per prompt. We calculated the score for each sample using the current PRM version and filtered completions with a score lower than the average for that prompt. We then applied variance-based filtering, selecting samples with $z > 1.96$.

**2. RL Post-Training (Using Trained PRMs for RL)** We evaluated the utility of our trained PRMs by using them as reward signals for reinforcement learning.

- **Model:** `Gemma-3-12B`, supervised by the trained PRMs in addition to a final correctness reward.

- **Algorithm:** Group Relative Policy Optimization (GRPO).

- **LoRA Configuration:**

  - Rank $r = 8$, Alpha $\alpha = 16$.
  - **Target Modules:** All linear layers.

- **Hyperparameters:**

  - **Learning Rate:** $5 \times 10^{-7}$ (AdamW).
  - **Group Size:** 16.
  - **KL Coefficient:** $\beta = 0.01$.
  - **Clip Ratio:** $\epsilon = 0.2$.

- **Reward Formulation:** Since GRPO requires the starting point to be consistent within the group, we calculated the reward as the sum of the PRM probabilities across all reasoning steps.

## M. Examples of LLM-Assisted Writing

We use LLMs only for making the writing concise. We provide the LLM with the text and ask it not to change any information, just make it concise. We then manually verify the text and make changes to it. A few examples are given below:

| Original | LLM + Human Polishing |
|---|---|
| A future work is to extend the adversarial formulation to other game-theory settings. In this paper, we focused on a two-player setup between a generator and a reward model, but multi-agent extensions could allow for diverse adversarial roles that challenge models along different dimensions of reasoning. Additionally, while our analysis showed convergence under OGDA with regularization, more work is needed to fully understand stability in large-scale adversarial training for language models. Exploring optimization techniques that reduce training instability while retaining the benefits of adversarial training is an important next step. | Future work includes extending the framework beyond the two-player formulation. Multi-agent variants could introduce diverse adversarial roles that probe different aspects of reasoning. On the optimization side, while our analysis demonstrated convergence with OGDA and regularization, further study is needed to understand stability at scale and to design algorithms that improve training efficiency while preserving the benefits of adaptive curricula. |

Another example:

| Original | LLM + Human Polishing |
|---|---|
| To access the effectiveness of our framework, we evaluate the accuracy of LLM solvers guided by our trained PRM against the strongest existing methods (See Table 1). Averaged across five math-reasoning benchmarks, under equivalent test-time compute budgets, `APRM` improves solver accuracy by ... over the strongest trained PRM baseline (ReST-MCTS) and by ... over best prompting methods (Tree-of-Thought). We observe the highest gains on OlympiadBench, which is an out-of-distribution (OOD) benchmark for the PRM. Here, `APRM` yields a substantial ... pp improvement over the best baseline (ReST-MCTS), which strongly indicates enhanced generalization and reduced overfitting. | Table 1, shows the performance LLM solvers guided by our PRM against prior methods under equivalent test-time compute. Averaged over five benchmarks, `APRM` improves solver accuracy by $+x$ pp relative to the strongest trained PRM baseline (ReST-MCTS (Zhang et al., 2024)) and by $+x$ pp relative to the best prompting method (Tree-of-Thought (Yao et al., 2023)). We observe the largest gains on *OlympiadBench*, which is an out-of-distribution benchmark, where `APRM` outperforms ReST-MCTS by $+x$ pp, suggesting better generalization beyond the training distribution. |

