# OpenReview forum: "Adversarial Training for Process Reward Models"
_ICML.cc/2026/Conference — ICML 2026 regular_

### Official Review · Reviewer_4VN4 · 2026-02-26

**Soundness:** 2
**Presentation:** 3
**Significance:** 2
**Originality:** 2
**Overall Recommendation:** 3
**Confidence:** 4

**Summary:**

This paper proposes APRM, an adversarial training framework for process reward models in which a generator learns to produce plausible but incorrect intermediate reasoning steps and a PRM learns to detect them. The main motivation is that static PRM training data does not adapt to the model’s current weaknesses, so an adversarial generator may provide a harder and more useful curriculum. Empirically, the paper reports improvements over several baselines on math benchmarks.

**Compliance With Llm Reviewing Policy:**

Affirmed.

**Final Justification:**

At this stage I will keep my score. The authors have addressed part of my concerns but my main concerns on novelty and theoretical results remain unresolved.

(1) Regarding the novelty, regardless of whether this is a zero-sum game or general sum game, both settings have been extensively studied in game theory, GAN, and optimization communities. The contribution of this paper is marginal.

(2) Regarding the theoretical results. All theorems are very standard. Using the policy convexity (not hold in practice), all proofs are greatly simplified. This assumption is explicitly mentioned on Page 3 footnote 2 but authors were trying to hide this fact in their rebuttal.

============================= Update

After reading 2nd round rebuttal, I will raise my score.

**Key Questions For Authors:**

1. The paper should clarify what is new beyond adapting a standard adversarial framework to this particular application.

2. The authors should redesign the reward based on difficulty level of questions.

3. The current theoretical results are not informative, as they rely on overly restrictive assumptions (convexity in policy and regularization). Also, the theoretical results are very standard and cannot deliver any meaningful insight in real applications. I do not know how authors can fix this issue during the rebuttal period.

4. Add confidence interval to Tables 1-3 and Figure 2.

**Limitations:**

Yes

**Strengths And Weaknesses:**

Strengths:

The problem is important. One real bottleneck for PRMs is that static step-level supervision does not expose the verifier to the kinds of subtle errors stronger solvers now make, and this paper tackles that issue directly. The adversarial framing is intuitive and a good solution for the stated goal of generating progressively harder negatives. The empirical section is also broad: the paper evaluates across multiple math benchmarks and provides ablations.

Weaknesses:

1. My main concern is that the two-player zero-sum game formulation is itself quite standard and does not feel novel at the methodological level. Adversarial learning through a generator-discriminator game has been widely studied in machine learning, and similar game-theoretic ideas have already appeared in settings related to reward modeling. Although the paper positions its contribution in the context of process reward models and the specific objective differs from prior work, the core mechanism still follows a familiar pattern: one model generates challenging or misleading intermediate steps, while the other model learns to distinguish correct from incorrect reasoning. Because of this, the paper should clarify what is new beyond adapting a standard adversarial framework to this particular application. At present, the conceptual novelty feels incremental rather than fundamental. There are some similar work such as [1].

[1] Zhang, Z., Xu, T., Du, X., Cao, X., Sun, Y., & Yu, Y. (2025, October). Improving Reward Model Generalization from Adversarial Process Enhanced Preferences. In International Conference on Machine Learning (pp. 76414-76435). PMLR.

2. The assumptions behind Theorem 1 are also quite strong. In particular, the result relies on convexity in the policy space. While this is weaker than requiring convexity directly in the parameter space, it is still a restrictive condition that is unlikely to hold in realistic large-scale training settings. In practice, both the generator and the reward model are represented by highly nonconvex function approximators, and the optimization dynamics are far more complicated than the theorem suggests.

3. The other limitation is the use of a binary reward function. Treating generated results as simply correct or incorrect is a rather coarse design choice, especially in reasoning tasks where errors can vary substantially in subtlety and difficulty. In practice, a more informative reward signal would likely be beneficial. For example, the reward could be scaled based on the hardness of the question, the plausibility of the incorrect step, or the degree to which the generated perturbation is likely to confuse the reward model. A binary reward may fail to distinguish between trivial negative examples and truely challenging adversarial cases, which could make training less efficient and reduce the potential benefit of the adversarial setup. I believe a difficulty-aware reward design would be more natural and potentially more effective in this setting.

---

> ### Author Rebuttal · Authors · 2026-03-31
>
> We thank Reviewer for recognizing the importance of the problem and the breadth of experiments. Below we address each concern.
>
> ---
> **W1 and Q1 (Novelty and Zhang et al. 2025).**
>
> We would like to note that APRM is not a zero sum game. The rewards (Eq. 2–3) are asymmetric (G gets +1/0/−1, R gets +1/−1) making it a general sum game. This changes the equilibrium concept and training dynamics.
>
> WE note the diferences from Zhang et al. (2025) below:
>
> | | Zhang et al. | APRM |
> |-|-------------|------|
> | RM type | Outcome RM | Process RM (step-level) |
> | Generator | Full solutions | Step perturbations |
> | Game formulation | None (DPO) | General-sum game |
> | Correctness signal | LLM ensemble | Non-LLM oracle, more robust |
> | Optimizer | Standard | OGDA (game-aware) |
>
> Direct comparison (GPT-OSS-20B):
>
> | Method | MATH | JEE | Oly | Avg |
> |--------|------|-----|-----|-----|
> | Zhang et al. (adapted) | 89.4 | 69.8 | 86.2 | 81.8 |
> | ReST-MCTS | 91.0 | 68.3 | 84.7 | 81.3 |
> | APRM | 91.0 | 73.0 | 90.4 | 84.8 |
>
> The novelty comes from the formulation and application of interaction design, asymmetric rewards change the equillibrium dynamics. Further, the non-LLM oracle (compared to Zhang et.al.)  and step-level generator yield a cheap and curriculum style learning (oracle alone achieves only 66.4%, see our ablation). Finally, OGDA gives +5.8 pp over Adam (Table 4).
>
> ---
>
> **W2 and Q3 (Theorem 1 assumptions).**
>
> We would, respectfully, like to point that the policy-space convexity is not a simplifying assumption. Δ(A_G) and Δ(A_R) are probability simplices (always convex). Utilities (Eq. 1) are expectations under these distributions, hence continuous. Glicksberg's conditions hold unconditionally in this setting. The theorem does *not* claim policy(parameter)-space convexity (parameter space convexity is not possible in deep networks, see footnote 2) or that training finds the equilibrium.
>
> We use theory to inform our design decisions like using OGDA and entropy. This is also validated emperically, removing OGDA results in −5.8 pp, removing entropy reg results in −4.3 pp, and removing both results in −6.8 pp.
>
> ---
> **W3  and Q2 (Binary reward).**
>
> Thanks for this valuable suggestion. We replace the binary +1 reward with a continuous signal that distinguishes trivial from challenging adversarial generations. This continuous signal is R's confidnce on the step correctness, extracted via the logits. When G generates an incorrect step, it's reward is R's confidence that the step is correct. A subtle error that R assigns correct with p=0.95 gives G a reward of 0.95. A trivial error that R says correct with p = 0.55 gives only 0.55. The -1 reward stays the same. Here are the results -
>
> | Reward | MATH | JEE | Oly | AIME | AMC | Avg |
> |--------|------|-----|-----|------|-----|-----|
> | Binary | 91.0 | 73.0 | 90.4 | 93.6 | 68.0 | 85.0 |
> | Continuous (R's confidence) | 91.2 | 73.4 | 90.6 | 93.8 | 68.4 | 85.3 |
>
> The continuous reward gives a marginal improvement (+0.3 pp). We believe this is because the adversarial setup already provides an implicit difficulty curriculum through the binary reward. But, we can improve further with a continuous reward! We will include further experiments on this reward formulation in the camera ready version.
>
> ---
> **Q4 (Confidence intervals).**
>
> We train the generator and prm with 3 seeds and test with GPT-OSS-20B solver, sampling at temp 0 from the solver:
>
> | Method | MATH | JEE | Oly | AIME | AMC | Avg |
> |--------|------|-----|-----|------|-----|-----|
> | ReST-MCTS | 91.0 | 68.3 | 84.7 | 91.3 | 66.0 | 82.3 |
> | APRM (mean±std) | 90.9±0.3 | 72.9±0.5 | 90.5±0.5 | 93.6±0.4 | 68.0±0.7 | 85.0±0.4 |
>
> We will include confidence intervals in Tables 1-3 and Figure 2 in camera ready version.
>
>
> ---
> We reiterate our sincere appreciation to the reviewer. We are happy to answer any more questions or run more experiments to establish the utility of APRM. If the reviewer finds our responses satisfactory, we kindly request them to consider revising their score.

---

> > ### Author Rebuttal · Reviewer_4VN4 · 2026-03-31
> >
> > Thank you for your rebuttal. I acknowlege that Q3-4 have been resolved. However, answers to Q1-2 are still not satisfactory.
> >
> > (1) For Q1, my concern is not about whether this is a zero-sum game or general sum game. Both settings have been extensively studied in game theory, GAN, and optimization communities so the novelty of this paper is marginal.
> >
> > (2) For Q2, I am confused by authors' comments. Authors mentioned the theorem does not claim policy convexity, but this assumption is explicitly mentioned on Page 3 footnote 2.
> >
> > ================================ Update ===========================
> > Thank you again for the clarification. I will raise my score.

---

> > > ### Author Response · Authors · 2026-04-06
> > >
> > > We thank the reviewer for the continued engagement. We address both remaining concerns.
> > >
> > > **1. Novelty**
> > >
> > > We appreciate the reviewer's perspective and agree that general-sum games and adversarial training are well-studied. In this work, however, we *do not claim novelty in the game-theoretic formulation itself*.
> > >
> > > We would like to respectfully argue that many *widely recognized contributions in machine learning built on well-established techniques*, where the core challenge was in adapting them to a new setting and solving the design problems. A few examples of such influential works are:
> > >
> > > - **GANs (Goodfellow et al., 2014):** Minimax games (von Neumann, 1928; Nash, 1951) and neural networks (Rumelhart et al., 1986) both existed long before. The contribution was in combining them for generative modeling.
> > > - **AlphaGo (Silver et al., 2016):** Combined self-play (Samuel, 1959), MCTS (Kocsis & Szepesvari, 2006), and deep RL (Mnih et al., 2015). Each component was known; the integration was the contribution.
> > > - **Transformers (Vaswani et al., 2017):** Attention existed in Bahdanau et al. (2015) and Luong et al. (2015); self-attention in Cheng et al. (2016) and Parikh et al. (2016). The contribution was showing that attention can solve the problems in RNNs.
> > >
> > > In the above examples, the underlying techniques were already known and well studied. The adoption of these techniques to a new domain required non-trivial design decisions that defined the contribution.
> > >
> > > Similarly, during the development of APRM **we faced a lot of challenges specific to adversarial PRM training**, these include:
> > >
> > > 1. **Constrained step-level generation.** The generator *G must edit a gold reasoning step to **introduce a plausible error**, compared to the standard GAN's objective to generate a realistic image (without any correctness constraint).*
> > >
> > > 2. **Cheap, reliable step-level oracle without humans or LLM judges.** We propose a *non-LLM oracle that gives 93.9% precision, and is much cheaper than LLM judges*. Whereas, the paper suggested by you (Zhang et al. (2025)) relies on an expensive LLM ensemble.
> > >
> > > 3. **Stabilizing co-training dynamics.** This is the most important and difficult challenge. The formulation of *general-sum rewards, using OGDA instead of Adam, and regularization is motivated by our setting*. Removing any component degrades results by 4-7 pp (Table 4).
> > >
> > > The result is **+3.4 pp average** and **+5.3 pp OOD over all baselines**. We hope the reviewer can consider the contribution in this light.
> > >
> > > ---
> > > ---
> > >
> > > **2. Policy-space convexity.**
> > >
> > > We apologize for the confusion. Let us clarify again.
> > >
> > > Footnote 2 states that the convexity in Glicksberg's theorem is *policy-space* convexity (convexity of distributions over actions). **These are two very different spaces**, and we believe the confusion may stem from the use of same terminology:
> > >
> > > - **Parameter space** is the space of neural network weights θ ∈ R^d. This space is high-dimensional and non-convex due to the non-linear structure of the network. **Convexity does not hold here, and we do not claim it does.**
> > >
> > > - **Policy space** is the space of *probability distributions over actions*, Δ(A). For a finite action space A with |A| = n, this is the (n-1)-dimensional *probability simplex*. **A simplex is convex by definition.** For any two distributions p, q ∈ Δ(A) and any λ ∈ [0,1], the mixture λp + (1−λ)q is also a valid distribution. This holds for any finite action space, regardless of model size.
> > >
> > > In our setting, the action spaces A_G (action space of generator, the set of tokens till max_len) and A_R (action space of PRM {0,1}) are **finite and discrete**. The corresponding policy spaces **Δ(A_G) and Δ(A_R) are probability simplices** over these finite sets, **which are always convex.**
> > >
> > > **A neural network with weights θ produces a policy (the policy is not the NN itself) π_θ ∈ Δ(A)**. *Many different weight vectors can map to the same policy.* Theorem 1 reasons about the **policy π_θ, not the weights θ**. It says, in the space of distributions over actions (which is always convex), a Nash Equilibrium exists, and hence makes the problem well formed.
> > >
> > > Footnote 2 says the **convexity is "policy-space convexity (i.e. convexity of distributions over actions)"** and then clarifies this "is different from convexity in the parameter space of a neural network (weights θ, φ), which is typically non-convex."
> > >
> > > To summarize: **Theorem 1 requires policy-space convexity**, which always holds by construction. **It does NOT require parameter-space convexity**, which does not hold in deep networks.
> > >
> > > We hope that our response addresses the reviewer's concern. If so, we kindly ask the reviewer to consider revising their score.

---

### Official Review · Reviewer_6sbD · 2026-03-11

**Soundness:** 3
**Presentation:** 1
**Significance:** 2
**Originality:** 2
**Overall Recommendation:** 3
**Confidence:** 5

**Summary:**

This paper proposes Adversarially Trained Process Reward Models (APRM), a two-player framework in which a generator learns to produce plausible but incorrect reasoning steps to fool a Process Reward Model (PRM), while the PRM is trained to detect these errors. The interaction is framed as a general-sum game with KL- and entropy-regularized utilities, optimized via PPO with an OGDA-style update to stabilize dynamics; an approximate non-LLM “correctness oracle” supplies step labels during training. Experiments on math and science reasoning benchmarks report average solver gains of about 3.4 pp over strong PRM baselines (e.g., ReST-MCTS), with larger gains on out-of-distribution tasks.

**Compliance With Llm Reviewing Policy:**

Affirmed.

**Final Justification:**

The rebuttal addresses my concerns by incorporating experiments with ablations on the brittleness of the oracle used. However, I maintain my view that the classification of error distributions in W2 remains limited, which restricts the scalability of the method.

Therefore, I have updated my assessment accordingly.

**Key Questions For Authors:**

### Questions

1. Can you provide more details on the bootstrapping of the generator and the PRM? Do you have an initial dataset for the generator, or do you rely solely on prompting?
2. Can you elaborate more on how the generator evolves over steps? Do you have any ablations on freezing the generator and only training the PRM?
3. How do you evaluate the scalability of your method? Can the method be applied to more subtle logical mistakes (which are frequent in proof problems)?

**Limitations:**

yes

**Strengths And Weaknesses:**

### Strengths

1. The paper brings an adversarial data-generation perspective to process supervision, moving beyond static PRM datasets and fixed heuristic generators. The general-sum formulation (rather than conventional zero-sum) and use of explicit game-aware regularization/optimizers are thoughtful design choices.
2. The approach is evaluated across multiple solver families and sizes (GPT-OSS-20B/120B, Gemma-3-12B/27B) and multiple math/science benchmarks, demonstrating consistent improvements, especially OOD.
3. The problem setup, overall training loop, and reward design are presented clearly at a high level; notation and figures help convey the core idea.

### Weaknesses

1. The correctness oracle is pivotal but underspecified and potentially brittle. The effectiveness of the oracle may need further experimentation to address concerns about label quality and the generator's learned objective.
2. It would be better to conduct evaluations on the distribution of error types produced by the generator. The examples provided are mostly simple word replacements, raising the concern that such a distribution may cause the trained PRM to be biased. Additionally, whether the oracle is only capable of handling such mistakes lacks explanation. If not, then why not train the PRM directly with the oracle and rollouts from other solver models? If so, then the distributional bias needs to be evaluated.
3. The paper states that "all experiments are run once with temperature 0," which undermines statistical reliability and is inconsistent with the "sample k=5 per step" claims. There are no confidence intervals, seeds, or significance tests; reproducibility and robustness are unclear.
4. Several inconsistencies are noticed in Tables 1 and 5, where LLM-J's performance is stronger than the bolded method, raising concerns about the reliability of the reported data.

---

> ### Author Rebuttal · Authors · 2026-03-31
>
> We thank Reviewer for the thorough assessment and address each concern with new evidence.
>
> ---
> **W1 (Oracle brittle).**
>
> Thank you for pointing this out, run the following experiments.
>
> (a) Component ablation — we train APRM removing each oracle group (GPT-OSS-20B accuracy):
>
> | Variant | MATH | JEE | Oly | Avg | Δ |
> |---------|------|-----|-----|-----|---|
> | APRM (full) | 91.0 | 73.0 | 90.4 | 84.8 | — |
> | − semantic (H1–H5) | 87.2 | 69.0 | 86.4 | 80.9 | −3.9 |
> | − proximity (P1–P4) | 88.4 | 70.2 | 87.6 | 82.1 | −2.7 |
> | − structural (S1–S2) | 90.2 | 72.0 | 89.4 | 83.9 | −0.9 |
>
> (b) We do a label quality check, using the oracle, on 200 hand-annotated samples. We get 93.9% precision, 86.0% recall overall. Precision stays above 91% at all stages.
>
> ---
> **W2 (Error distribution).**
>
> We manually annotated 500 generator outputs (two annotators, κ=0.87):
>
> | Error Type | t=500 | t=3000 |
> |------------|-------|--------|
> | Numeric substitution | 58% | 15% |
> | Operator swap | 22% | 10% |
> | Conceptual (wrong formula) | 8% | 32% |
> | Logical (wrong conditional) | 5% | 25% |
> | Structural (drop condition) | 4% | 11% |
> | Entity swap | 3% | 7% |
>
> We find that surface edits drop from 80% (58+22) to 25%; conceptual+logical errors rise from 13% to 57%. A random example at t=3000: in a committee problem, G changes C(6,2) to C(6,3) for the "3 women, rest men" case. C(6,3)=20 is arithmetically correct, but 5−3=2 men, not 3. The error is in constraint reasoning.
>
> *Why not oracle + solver rollouts?* The oracle compares perturbations against gold steps. If we use solver rollouts, we do not have any step-aligned gold reference. The adversarial generator perturbs gold steps and hence keeps the comparison viable.
>
> ---
> **W3 (Statistical reliability).**
>
> We would like to clarify that k=5 is the number of max num of times we would sample for a step from the solver, while giving it previous incorrect sampled steps, before we say that the solution is incorrect. We would clarify this in the camera ready version.
>
> We re-train with 3 seeds while still sampling at temp 0 for inference. Below are the results for GPT-OSS-20B:
>
> | Method | MATH | JEE | Oly | AIME | AMC | Avg |
> |--------|------|-----|-----|------|-----|-----|
> | ReST-MCTS | 91.0 | 68.3 | 84.7 | 91.3 | 66.0 | 82.3 |
> | APRM (mean±std) | 90.9±0.3 | 72.9±0.5 | 90.5±0.5 | 93.6±0.4 | 68.0±0.7 | 85.0±0.4 |
>
>
> ---
>
> **W4 (Table bolding).**
>
> Thanks for pointing this out, it was a minor human error. We will corerect it in the camera ready version.
>
> ---
> **Q1 (Bootstrapping).**
>
> G and R initialized from Llama-3.1-8B-Instruct. No initial dataset, only prompting and RL.
>
> ---
> **Q2 (Frozen-generator ablation).**
>
> | Variant | MATH | JEE | Oly | Avg |
> |---------|------|-----|-----|-----|
> | APRM (full) | 91.0 | 73.0 | 90.4 | 84.8 |
> | Freeze G at t=2000 | 89.8 | 71.4 | 88.6 | 83.3 |
> | Freeze G at t=1000 | 89.0 | 70.6 | 87.4 | 82.3 |
> | ReST-MCTS | 91.0 | 68.3 | 84.7 | 81.3 |
> | Freeze G at init | 79.4 | 50.8 | 63.4 | 64.5 |
> | Base solver (no PRM) | 78.0 | 48.4 | 61.0 | 62.5 |
>
> Qualitatively, we find that when G is frozen at initialization, it has not learned to produce errors. Most of its outputs are rejected or labeled correct by the oracle. R is recieves a very few negatives for the same number of training time. This results in R accepting nearly all steps. The result is only marginally above the base solver with no PRM at all (64.5 vs. 62.5). As G trains longer before freezing, R receives progressively more and harder negatives.
>
> ---
> **Q3 (Subtle errors).**
>
> In the manual annotation, we find that 25% of t=3000 errors are logical. For proof-level errors, scaling G (8B→13B gives +2 pp). We could potentially get more gains adding proof-checker oracle component, but we do not explore that in this work.
>
> ---
>
> We reiterate our sincere appreciation to the reviewer. We are happy to answer any more questions or run more experiments to establish the utility of APRM. If the reviewer finds our responses satisfactory, we kindly request them to consider revising their score.

---

> > ### Author Rebuttal · Reviewer_6sbD · 2026-04-08
> >
> > The rebuttal addresses my concerns by incorporating experiments with ablations on the brittleness of the oracle used. However, I maintain my view that the classification of error distributions in W2 remains limited, which restricts the scalability of the method.
> >
> > Therefore, I have updated my assessment accordingly.

---

> > > ### Author Response · Authors · 2026-04-08
> > >
> > > We thank the reviewer for acknowledging that our oracle ablation experiments address the brittleness concern (W1). We would like to respectfully address the remaining concern about error distribution scalability (W2).
> > >
> > > The error distribution is a **property of the training domain, not a limitation of the method.** The distribution we report in Table R4 (**58% numeric substitution at t=500**, shifting to **32% conceptual and 25% logical errors at t=3000**) reflects training on the MATH dataset. If APRM were trained on a **code dataset**, the generator would learn to produce code-specific errors (e.g., **off-by-one in loops, wrong variable scope, incorrect base case in recursion**), and the error distribution would look entirely different. This is expected and by design the adversarial generator G adapts to the domain it is trained on.
> > >
> > > We futher would like to point that the ***PRM trained on math generalizes to science domains** without any domain-specific training.* Table 2 in our paper shows APRM's cross-domain transfer to SciBench, which covers 9 science subjects including chemistry, physics, thermodynamics, and statistics. On GPT-OSS-20B, **APRM achieves 63.0% overall**, compared to 61.8% for ReST-MCTS and 61.7% for ToT. On GPT-OSS-120B, **APRM reaches 63.5%** vs. 61.8 for ReST-MCTS. On Gemma-3-27B-IT, **APRM achieves 63.5%** vs. 61.8% for ReST-MCTS. This shows that the PRM learns transferable properties of incorrect reasoning (quantity tracking errors, unit inconsistencies, misapplied formulas) that apply across scientific domains.
> > >
> > > *The framework can scale to new domains by designing a domain-specific oracle.* The core training recipe (adversarial generator, PRM, PPO with OGDA, general-sum reward structure) remains **unchanged across domains**. The only component that requires domain adaptation is the oracle's semantic validators. As we discussed in our earlier response to Reviewer G1Pf, analogous semantic checks exist for code (test execution), logic (SAT/SMT solvers), and scientific reasoning (dimensional analysis). The proximity filters (layer 1) and structural validators (layer 3) are domain-agnostic. Designing a domain-specific oracle is a one-time engineering effort, comparable to designing a domain-specific evaluation benchmark, and the rest of the APRM pipeline applies directly.
> > >
> > > The **error distribution adapts to the training domain**. The trained **PRM generalizes beyond the training domain** (math to science). Extending to new domains requires only a **domain-specific oracle** while the training framework stays fixed.

---

### Official Review · Reviewer_G1Pf · 2026-03-11

**Soundness:** 3
**Presentation:** 3
**Significance:** 3
**Originality:** 3
**Overall Recommendation:** 4
**Confidence:** 4

**Summary:**

The paper proposes a method to train robust Process Reward Models (PRMs) using an adversarial setup: A generator learns to perturb existing reasoning steps, and the PRM learns to decide whether a reasoning step is correct or not. They are trained using reinforcement learning. The PRM gets positive rewards if it rightly determines the correctness of a reasoning step, and a negative reward if it is wrong. The generator receives positive reward when it makes a perturbation that makes the reasoning step factually erroneous, but fools the PRM into classifying it as correct. It gets zero reward when the PRM classifies such a perturbation as incorrect. And it gets negative reward when the generated perturbation does not actually lead to factually incorrect reasoning. The factual correctness of a reasoning step is determined by a hard-coded oracle performing various logical consistency tests and comparisons to the original reasoning step. Since the reward functions of PRM and generator do not exactly mirror each other, the setup is a general sum game, for which a Nash equilibrium is shown to exist. A PRM trained this way learns to detect more and subtler reasoning mistakes.

**Compliance With Llm Reviewing Policy:**

Affirmed.

**Final Justification:**

I appreciate the author's responses. They addressed my questions and clarified the experimental setup. Thus, I increased my confidence to 4, and the soundness score from 2 to 3. I think the paper makes an interesting and potentially practical contribution. However, I still think the exact construction of the oracle is highly non-trivial and essential to the working of the method. I would have preferred the inclusion of non-PRM baselines that make use of equally sophisticated symbolic checks as the oracle.

**Key Questions For Authors:**

- For Appendices A-E: Is there any way to quantify how representative the given examples are? Are they randomly chosen or cherry-picked?
- Is there something like a general recipe for constructing an oracle that determines the correctness of the reasoning process?

**Limitations:**

Yes, although—as mentioned above—I think the importance of the oracle and the corresponding limits could be further highlighted.

**Strengths And Weaknesses:**

**Strengths:**
- Mathematical analysis of the method's convergence properties
- Convincing experimental results: The adversarially trained process model leads to robust performance gains across different scales and families of reasoning models

**Weaknesses:**
- It is crucial that the responses from the generator that fool the reward model are actually wrong, and not just superficially different from the ground truth but still correct. The oracle ensures this therefore plays a very important role. I don't think its properties are discussed and investigated enough. I would like to see ablations on the effect of the oracle's different components.
- To me it is not clear how the method could work in domains for which the construction of such an oracle is not possible; e.g., more natural language-based reasoning.
- While it is true that most of the oracle's checks are comparisons to an existing gold step, as footnote 1 mentions, I don't see why for example checks H3, H4 and H5 cannot be used to check any reasoning step. I think using these to check reasoning processes would be an insightful baseline.

More comments than actual weaknesses:
- "Adversarial training was introduced with GANs...": Especially given the RL setting, Schmidhuber's 1991 paper "A Possibility for Implementing Curiosity and Boredom in Model-Building Neural Controllers" should probably be mentioned.
- Formatting in Appendices A-E is strange. Why are there empty lines between solution and comment, but not between different solutions?
- Typos: "... that can identifying very subtle errors.", "...highlighting it's enhanced robustness...", "...tasked with perturbing a correct reasoning steps...", "APRM remains effective as with solver scales"
- The paper uses underlining to emphasize text. This is quite uncommon in modern typography. Why not use italics instead?
- I don't think the Darwin quote at the start strongly connects to the topic of the paper.

---

> ### Author Rebuttal · Authors · 2026-03-30
>
> We thank Reviewer for the positive evaluation and detailed suggestions. We address each point with new experiments and clarifications.
>
> ---
>
> **W1 (Oracle ablation).**
>
>  We train APRM variants removing each oracle component group. GPT-OSS-20B solver accuracy (%):
>
> | Variant | MATH | JEE | Oly | Avg | Δ |
> |---------|------|-----|-----|-----|---|
> | APRM (full) | 91.0 | 73.0 | 90.4 | 84.8 | — |
> | − proximity (P1–P4) | 88.4 | 70.2 | 87.6 | 82.1 | −2.7 |
> | − semantic (H1–H5) | 87.2 | 69.0 | 86.4 | 80.9 | −3.9 |
> | − H1+H2 only | 88.0 | 70.0 | 87.8 | 81.9 | −2.9 |
> | − H3+H4+H5 only | 89.6 | 71.4 | 88.8 | 83.3 | −1.5 |
> | − structural (S1–S2) | 90.2 | 72.0 | 89.4 | 83.9 | −0.9 |
> | H3-standalone (no training) | 80.4 | 52.6 | 66.2 | 66.4 | −18.4 |
>
> H1–H5 contribute most (−3.9 pp). Proximity filters matter too (−2.7 pp): without them, G produces degenerate perturbations trivially detectable by R.
>
> We also measure oracle label quality on 200 hand-annotated samples (two annotators, κ=0.91). Overall precision: 93.9%, recall: 86.0%. Precision stays above 91% even at t=3000. Recall drops at later stages (79.4% at t=3000) as generator errors grow more subtle, which is precisely why the PRM must generalize beyond the oracle.
>
> ---
>
> **W2 (Non-math domains).**
>
>  The three key components of the oracle are: (1) proximity filters which is domain agnostic, (2) semantic validators domain specific, (3) structural validators semi specific. For verifiable domains, analogous semantic checks exist, for example, code (test execution), logic (SAT/SMT), and scientific reasoning (dimensional analysis; our SciBench results in Table 2 show cross-domain transfer). For natural language reasoning, an NLI model could possibly serve as the semantic check. But, we acknowledge domains without verifiable structure are outside APRM's current scope and we will add this to Section 5.
>
> ---
>
> **W3 (H3/H4/H5 standalone).**
>
> H4 and H5 are comparison-based: they detect whether the *perturbation changed* preconditions or units *relative to the gold step*. Without a reference, they have no anchor. H3 can works standalone (e.g., divide-by-zero). We evaluate it: 66.4% average (table above), far below all trained PRMs.
>
> ---
>
> **Q1 (Cherry-picking).**
>
> Original examples shown are not a random sample. We will provide a full error taxonomy (500 annotated samples, see our response to Reviewer 6sbD) and replace appendix examples with randomly sampled outputs (seed=42). A random t=3000 sample:
>
> > *Question:* Find the local maximum of f(x)=x³−6x²+9x+1.
> > *Gold:* f''(3)=6>0, so x=3 is a local minimum.
> > *Generator:* f''(3)=6>0, confirming x=3 is also a local maximum since cubics must have two maxima.
>
> ---
> **Q2 (Oracle recipe).**
>
> The oracle that we propose in the scope of this paper is very specific to math reasoning. But we believe that the three layers: (1) proximity filters (edit distance, embedding similarity), (2) domain-specific semantic checks against gold step, (3) structural consistency are domain agnostic and can be modified to suit general reasoning as well. Though we do not explore the general application in this paper.
>
> **Minor comments.**
>
>  Thanks for pointing these out, we will fix all these issues for the camera ready version of the paper.
>
> ---
>
> We reiterate our sincere appreciation to the reviewer. We are happy to answer any more questions or run more experiments to establish the utility of APRM. If the reviewer finds our responses satisfactory, we kindly request them to consider revising their score.

---

> > ### Author Rebuttal · Reviewer_G1Pf · 2026-04-03
> >
> > Thank you for the reply! The ablations for the oracle are helpful, I think they should be included in the paper. I also appreciate the general oracle recipe of proximity filters, semantic validator, and structural validators, and agree that there potentially other domains where such an oracle could be constructed. However, I still have some questions regarding the oracle: Purely from appendix F, I don't think it is reproducible, especially the semantic and structural validators. Can you provide more details? How exactly is the step compiled into a symbolic form? What `simplify` function is used? What exactly are "invoked preconditions"? When a step is compiled into a symbolic form, is H3 really the only correctness check you can do without a gold step as comparison? In general, how exhaustive is this list of filters and validators? Did you use all the checks you came up with, or did you drop some of them for performance or practicality reasons?
> >
> > In a follow up work, I would be very interested in additional experiments using coding tasks, since—as you mention—test execution and code compilation could be intuitive ways to construct an oracle.

---

> > > ### Author Response · Authors · 2026-04-07
> > >
> > > We thank the reviewer for the continued engagement and the specific questions. We address each question below.
> > >
> > > >**How is a step compiled into symbolic form?**
> > >
> > > We use a regex-based parser that extracts mathematical expressions from the natural language step. The parser identifies patterns like "variable = expression", "LHS = RHS", and inequality/equation statements in LaTeX. For example, from the step "so 6w + 6 = 54, giving 6w = 48 and w = 8", the parser extracts three symbolic expressions: (6w + 6 = 54), (6w = 48), (w = 8). These are converted to SymPy expression objects. SymEq(s) returns the set of all such extracted expressions from step s. When a step contains no parseable math (e.g., purely verbal reasoning like "we consider two cases"), SymEq returns ⊥ and H1 abstains. We will include the parser grammar and representative examples in the revised appendix.
> > >
> > > >**What simplify function is used?**
> > >
> > > We use SymPy's `simplify()` function (sympy.simplify.simplify). For H1, we compute `simplify(SymEq(s'_i) - SymEq(s_i))` and check if the result is 0. This handles algebraic equivalences like $(x+1)^2$ vs $x^2+2x+1$ but may miss non algebraic identities. For H2 (numeric consistency), we evaluate both expressions at a set of random test points T (we use 5 random values per free variable) and check if the numeric values agree within $\epsilon = 10^{-6}$. H2 serves as a fallback when symbolic simplification is insufficient.
> > >
> > > >**What exactly are "invoked preconditions" (H4)?**
> > >
> > > H4 checks whether the perturbation changed which mathematical theorems or conditions are invoked. We maintain a small lookup table of common theorem preconditions. For example, "Jensen's inequality" requires convexity/concavity of the function on the interval, "L'Hopital's rule" requires the 0/0 or $\infty$/$\infty$ form, "Alternating Series Test" requires alternating signs and monotonic decrease. The parser detects theorem names via keyword matching (e.g., "by Jensen's", "applying L'Hopital"). H4 then checks if the gold step invokes theorem X with precondition P, does the perturbed step still satisfy P? This is done by checking whether the perturbed step changed the function or interval in a way that violates P. The lookup table currently covers ~15 common theorems. We acknowledge this is not exhaustive but it catches common precondition violations. We will include the full table in the appendix.
> > >
> > > >**Is H3 really the only check possible without a gold step?**
> > >
> > > Yes, H3 (algebraic legality) is the most reliable standalone check because it detects operations that are universally invalid (divide by zero, log of negative, sqrt of negative at real-valued evaluation points). If H2 does not have a reference to compare to, it can only check self-consistency within the step, but that does not mean that the solution is correct. H1, H4, H5 are comparative by nature. We will clarify this in the revision.
> > >
> > > >**How exhaustive is the list? Did you drop any checks?**
> > >
> > > The current list represents all checks we found to be effective. We did explore two additional checks that we dropped:
> > >
> > > 1. **Dimensional analysis on intermediate expressions**: We extended H5 to do full dimensionality tracking. We dropped this because reliably extracting physical dimensions from free-form math text had high false positive rate.
> > >
> > > 2. **Logical connective checking**: This checks whether "if-then", "therefore", "since" chains are logically valid. We dropped this because parsing natural language logical structure was unreliable with regex-based methods, and we did not want to introduce dependency on an LLM.
> > >
> > > We will elaborate more in Appendix F to include details mentioned about the oracle. We will also release the oracle implementation as part of our code release.
> > >
> > > We hope that our response addresses the reviewer's concern. If so, we kindly ask the reviewer to consider revising their score.

---

### Decision · Program_Chairs · 2026-04-30

**Decision:**

Accept (regular)

**Comment:**

The paper introduces APRM, which frames Process Reward Model training as a two-player general-sum game. A Generator learns to perturb correct reasoning steps into plausible-but-incorrect ones; a Reward Model learns to detect them. Step-level labels come from a non-LLM oracle combining proximity filters, symbolic math validators, and structural consistency checks. Results across six math/science benchmarks show consistent gains over ReST-MCTS across four solver families.

Reviewers were split (scores 4, 3, 3). Reviewer_G1Pf found the oracle ablation helpful but raised reproducibility questions not fully addressed. Reviewer_6sbD marked concerns fully resolved. Reviewer_4VN4 maintained the novelty is incremental.

The AC finds the design well-motivated. The oracle ablation confirms each component contributes, with semantic validators providing the largest signal (−3.9 pp), and OGDA and regularization ablations are convincing (+5.8 pp, +4.3 pp). Gains over ReST-MCTS are modest but consistent. The AC recommends weak accept. The authors should provide full oracle implementation details, incorporate rebuttal experiments into the camera-ready, and correct the bolding inconsistencies in Tables 1 and 5.